



# Quantifying processes contributing to coastal hazards to inform coastal climate resilience assessments, demonstrated for the Caribbean Sea

Svetlana Jevrejeva [1,2] , Lucy Bricheno[1], Jennifer Brown[1], David Byrne[1], Michela De Dominicis[1], Andy Matthews[1], Stefanie Rynders[1], Hindumathi Palanisamy[2] and Judith Wolf[1]

[1]National Oceanography Centre, Liverpool, L3 5DA, the UK
[2]Centre for Climate Research Singapore, Singapore, Singapore

*Correspondence to*: Svetlana Jevrejeva (Svetlana.jevrejeva@gmail.com)

**Abstract.** Scientific evidence is critical to underpin the decisions associated with shoreline management, to build climate resilient communities and infrastructure. We explore the role of waves, storm surges and sea level rise for the Caribbean region with a focus on coastal impacts in the eastern Caribbean islands. We simulate past extreme events and a worst-case scenario, modelling the storm surges and waves, suggesting a storm surge might reach 1.5m, depending on the underwater topography. Coastal wave heights up to 12m offshore and up to 5m near the coast of St Vincent are simulated with a regional wave model. We deliver probabilistic sea level projections for 2100, with a low probability/high impact estimate of possible sea level rise up to 2.2m, exceeding the 1.8m global estimate for the same scenario.

We introduce a Combined Vulnerability Index, which allows a quantitative assessment of relative risk across the region, showing that sea level rise is the most important risk factor everywhere, but wave impacts are important on windward coasts, increasing to the north, towards the main hurricane track. Our work provides quantitative evidence for policy makers, scientists, and local communities to actively prepare for and protect against climate change.

## 1        Introduction

Climate change is happening worldwide – no region on Earth has escaped it (IPCC, 2014). Caribbean small island states are particularly vulnerable to coastal climate change since the socio-economics of small islands rely on the preservation of the coastal zone. This presents a significant risk to the region's people (IPCC, 2014; Caribbean Marine Climate Change Report Card, 2017; CARIBSAVE, 2012). Small Island Developing State (SIDS), with a high concentration of population, infrastructure, and services in the low-lying coastal areas, are particularly exposed to rising sea levels, intense storms and coastal erosion. These are already posing severe threats to people (property, infrastructure and livelihoods, such as tourism and artisanal fisheries) and the marine and coastal ecosystems that support them (Rhiney, 2015). They can have a severe impact on the economy, in some cases over 100% GDP for a single event (Jevrejeva et al., 2018; Monioudi et al., 2018; Caribbean Marine Climate Change Report Card 2017; CARIBSAVE, 2012).





Governments of SIDS in the Caribbean recognise that climate change and sea level rise, in particular, are serious threats to the sustainable development and economic growth of the Caribbean islands. They recognise that urgent actions are required to
increase resilience and make decisions about how to adapt to future climate change (Caribbean Marine Climate Change Report Card, 2017; IPCC, 2014). Often Governments of SIDS are limited in resources to undertake these actions. The cost of inaction in the Caribbean alone is projected to amount to over US$22 billion annually by 2050 – equalling 10 percent of the current size of the Caribbean economy (United Nations, 2015; http://unohrlls.org/sids-in-numbers-climate-change-edition-2015/).

Although the level of vulnerability might vary from island to island, it is expected that practically all small island
states will be adversely affected by sea level rise, together with intermittent extreme events due to waves and storm surges. Storm surges are often quoted as causing coastal flooding but waves are often overlooked in terms of their coastal impact (Melet et al., 2016, 2018). Locally generated waves (wind-sea) or remotely generated swell can make significant contributions to coastal sea level changes through wave setup (due to wave dissipation through breaking and bottom friction) and runup/swash of individual waves (Stockdon et al., 2006). Compared to some other parts of the world, the potential impacts of
sea level rise on Caribbean islands have not been fully understood, due to: lack of sea level and wave observations; limited understanding of sea level rise and its variability; coastal processes and coastal geomorphology of the islands; and lack of detailed studies about sea level rise impact in coastal areas (Wolf, 1996).

In this study, we explore the role of waves, storm surges and sea level rise for the Caribbean region, with a focus on coastal impacts using St Vincent and the Grenadines (SVG) as an example. We address the urgent need to understand the
impact of a warming climate on the coasts of SIDS for decision-making about coastal adaptation. The majority of the infrastructure and settlements in SVG, like most SIDS, are located on, or near the coast, including government, health, commercial and transportation facilities. The main island of St Vincent has steep topography, which acts to concentrate development and transport closer to the coast – putting valuable assets and infrastructure more at risk. It also tends to increase the degradation of coastal and marine biodiversity, thereby reducing resilience to climate change impacts such as sea level
rise, waves and storm surge. High-density tourism development on the coast is particularly vulnerable to climate change and sea level rise, as are fish-landing centres.

Understanding the nearshore processes that form the key drivers for coastal impact is vital to define scenarios for impact assessments, however before the local impacts of climate change can be assessed, an understanding of the regional changes in water levels and waves must first be obtained. By quantifying the contributions to coastal hazards due to water
levels and waves, we define the nearshore conditions that need consideration in sensitivity assessments to explore plausible future change and build coastal climate resilience into land use plans and coastal infrastructure management plans. In section 2 we describe some of the key characteristics of the Caribbean Sea, the Lesser Antilles and St Vincent in particular. In section 3 we introduce the regional modelling tools to be used for hydrodynamic and wave modelling, together with model validation and the case study methodology. In section 4 we show the results for the model case studies and future sea level. Section 5





introduces the concept of a vulnerability index, combining the effects of tides, waves, water levels and storm winds. Section 6 is a discussion of the results.

## 2.    Study area: the Caribbean Sea and the Lesser Antilles

The Caribbean Sea is a marginal sea of the Atlantic Ocean in the tropics of the Northern Hemisphere. It is bounded by Mexico and Central America to the west and south west, to the north by the Greater Antilles, to the east by the Lesser Antilles, and to
the south by the north coast of South America. To the northwest the Caribbean Sea is connected to the Gulf of Mexico through the Yucatan Channel (see Fig.1). The Greater Antilles is a grouping of the four larger islands in the Caribbean Sea: Cuba, Hispaniola (containing Haiti and the Dominican Republic), Puerto Rico, Jamaica, plus the Cayman Islands. Here we focus on the Lesser Antilles of the eastern Caribbean: consisting of the Windward and Leeward Islands and Leeward Antilles (or Dutch Antilles).

SVG is an archipelagic state that forms part of the Windward Islands in the south-eastern part of the Caribbean. Located at 13°15' N and 61°15' W, it is neighboured by St Lucia to the north, Barbados to the east and Grenada to the south (Fig.1). Although SVG lies to the south of the main hurricane storm track, the islands are occasionally impacted by tropical storms, hurricanes and heavy rainfall events. Most recently, heavy rainfall during April 11-12, 2011 caused rivers to overflow and landslides in the north-eastern section of St Vincent. An assessment by the National Emergency Management Office of
SVG revealed that the sectors most affected were water and agriculture. Accelerated sea level rise is expected to increase the likelihood of the inundation of low-lying coastal areas, increase the salinity of surface and ground water and result in higher water tables. The impact of sea level rise is likely to exacerbate the damage caused by existing anthropogenic impacts, such as coastal pollution and over-fishing. Improving the management of biodiversity and fisheries will become increasingly important to the welfare of Vincentians and to the sustainability of the country's main economic activities – fishing, tourism and
agriculture.

Beaches and coastal ecosystems (including coral reefs, mangroves and sea grass beds) are particularly vulnerable to sea level rise, more intense storm surges and changes in waves (IPCC, 2014).  In the small islands of the Grenadines, protecting fisheries is important for maintaining healthy populations of herbivores and hence the resilience of coral reefs, as well as for safeguarding the sustainability of artisanal fisheries.

**2.1 Recent hurricane impacts in the Caribbean Sea, especially Lesser Antilles**

The Atlantic hurricane season runs from June 1st to November 30th and includes the Atlantic Ocean, the Gulf of Mexico and the Caribbean Sea. The Saffir-Simpson Hurricane Wind Scale is a 1 to 5 rating based on a hurricane's sustained wind speed, which is used for classification of hurricanes. Table SI1 shows the range of wind speeds (in m s$^{-1}$) associated with each category of hurricane, together with a description of the likely damage which will be experienced if such a category of hurricane makes
landfall. Below Category 1, a storm can be classified as a tropical storm. A total of 32 years (1986-2017) of tropical cyclone





events, which have occurred in the Caribbean Sea, have been extracted from the IBTrACS (International Best Track Archive for Climate Stewardship) database (Knapp et al., 2010a). This period was selected in order to provide a 30-year present-day climatology (1986-2015), to complement that extracted from the 37-year (1979-2015) global wave model runs from Bricheno and Wolf (2018). This period has been extended to include the disastrous hurricane season of 2017, which saw 12 North

Atlantic hurricanes, 5 of which had some presence in the Caribbean Sea along with Tropical Storm Bret which impacted Trinidad and Tobago in June. In August, hurricane Harvey caused some damage in Barbados, then Category 5 hurricanes Irma and Maria in September caused major damage to the Caribbean islands of Barbuda (and St Martin) and Dominica respectively.

Over the 30 years there were 122 storms, of at least Tropical Storm strength, which spent at least part of their life (storm) cycle in the Caribbean Sea. Quantitative assessment of the damage caused is available, especially for the most recent

events. Most damage is caused by direct impact of winds on infrastructure including the electricity supply and homes, together with heavy rainfall causing landslides and mudslides. Coastal flooding and erosion may occur, due to combinations of high water levels and waves, but it is not always clear whether the increase in water levels has been due to storm surge or wave effects, as the term 'storm surge' may be used loosely. In the narrative reports of these tropical storms, there are only 9 explicit mentions of storm surge during this 30-year period and 5 mentions of waves (SI). There are 5 references to the occurrence of

surges in the Lesser Antilles: 1 in Dominica (Luis, 1995), 1 in the USVI (Marilyn, 1995), 1 in St Lucia (Omar, 2008) and 2 in Antigua and Barbuda (Omar, 2008; Gonzalo, 2014). For waves there is 1 occurrence in Trinidad (Iris, 1995), while Omar (2008) also caused notable occurrences in Antigua and Barbuda and St Croix (US Virgin Islands).

## 3.     Modelling Approach

### 3.1 Sea level projections

Global sea level rise is an integral measure of warming climate (Church et al., 2013a; Jevrejeva et al., 2016), reflecting alterations in the dynamics and thermodynamics of the atmosphere, ocean and cryosphere as a response to changes in radiative forcing. The primary climate-related contributors to sea level rise are ice loss of land-based glaciers and ice sheets in Greenland and Antarctica; and the thermal expansion of the oceans (Church et al., 2013a). In addition, there is a non-climate contributor - changes in water storage on land due to groundwater mining and the construction of reservoirs (Church et al., 2013a).

In this study we provide probabilistic projections for future sea level rise for SVG, which deliver a scientific estimate of future sea level rise and address the challenges of adapting to sea level rise in the region. Probabilistic sea level projections deliver probability density functions (PDF) that are conditional upon emissions scenarios, which self-consistently project both likely values of mean sea level rise and the likelihood of high risk, low probability conditions such as those associated with rapid ice mass loss of Antarctica or Greenland. Those projections, explicitly labelled "probabilistic", not only include central

or "likely" ranges (Church et al., 2013b) but also the tails of these distributions (Jevrejeva et al., 2014; Kopp et al., 2017). Probabilistic projections also provide a summary assessment of the relevant uncertainties, which are consistent with some of





the decision frameworks used by coastal engineers for infrastructure design and land use planning (e.g., Hunter et al., 2013; Kopp et al., 2014; Jevrejeva et al., 2019).

Probabilistic sea level projection at the coast must also take into account the local vertical land movement caused by
long-term glacial isostatic adjustments and subsidence due to ground water extraction, urbanization, and river delta sedimentation rates (Kopp et al., 2014; Grinsted et al., 2015; Jevrejeva et al., 2016; Jackson et al., 2018). However, in this study we do not include any vertical land movement corrections due to the lack of information about local changes.

Future sea level will rise in SVG due to melting of land-based glaciers; ice mass loss from the ice sheets in Greenland and Antarctica; and from the thermal expansion of ocean waters (Jackson and Jevrejeva, 2016). We utilised outputs from our
previous studies (Jackson and Jevrejeva, 2016; Jevrejeva et al., 2016) by calculating spatial patterns of dynamic changes in sea surface height (SSH) and global average steric sea level change from 33 models in the Coupled Model Intercomparison Project Phase 5 (CMIP5). Spatial patterns of ice loss from glaciers and ice sheets are derived from present-day spatial attribution of terrestrial ice loss (Bamber and Riva, 2010). The land-water fingerprint is calculated using projected changes in land water storage (Wada et al., 2012). At each point on the global ocean, a putative sea level can be generated by random
sampling of the component PDFs and summing. Repeating this process 5000 times provides enough realisations of sea level to create the probability density function for sea level rise at each point on globe. Finally to account for glacial isostatic adjustment (GIA), we add the time-integrated global sea level field from the ICE 5G model (Peltier et al., 2015) to the sum of sea level components. We extract projections applicable to SVG for a location with coordinates 13.5° N and 61.5° W ([www.psmsl.org/cme](www.psmsl.org/cme)).

**3.2 Global wave model data for the Caribbean Sea**

Global wave model data has been taken from a long hindcast simulation described by Bricheno and Wolf (2018). A global configuration of WAVEWATCH III (WW3) was forced by ERA-Interim reanalysis winds for the period 1979-2015. The modelled hourly significant wave height ($H_S$) has been validated, by comparison with wave buoy data in the Caribbean Sea. The buoys used for validation are mapped in Fig. 1.
Fig. SI2 is a scatter plot of significant wave height from model data against buoy observations. This comparison shows that wave heights are well-reproduced by the model, with an average mean-squared error of 0.12m and correlation coefficient of 0.85. Detailed statistics for each of the seven buoy sites are shown in Table SI3. Larger wave events (above 4m $H_S$) however are systematically under-predicted by the model with respect to observed wave heights (this is a well-known feature of modelling extreme events, due to underestimation of the peak winds in the atmospheric forcing). Fig. 2 shows a
short time series of modelled and observed $H_S$ at buoy 42060, which illustrates the generally good agreement between model and observations, but the typical underestimation of extreme events, such as Hurricane Tomas in 2010.



### 3.3 Regional models for extreme events

Regional models of the Caribbean Sea have been setup as separate hydrodynamic and wave models, extending from 5-32°N and 5-100°W, with resolution 1/12° lat/lon. The bathymetry and extent are shown in Fig. 1.

### 3.3.1 Case studies

Two historical case studies were considered for numerical experiments: Hurricane Ivan (2004) and Hurricane Tomas (2010). Both were locally significant storms and are typical of the size and magnitude of storms seen in the region. Ivan caused catastrophic damage in Grenada as a Category 3 hurricane, heavy damage in Jamaica as a Category 4, and then severe damage in Grand Cayman, Cayman Islands, and the western tip of Cuba as a Category 5 hurricane. Tomas moved through the
Windward Islands and passed over St Lucia as a Category 1 hurricane. After reaching Category 2 status, Tomas quickly weakened to a tropical storm in the central Caribbean Sea, due to strong wind shear and dry air, but later regained hurricane status as it reorganized near the Windward Passage between the islands of Cuba and Hispaniola, north of Jamaica.

We use two types of atmospheric forcing in our numerical experiments: historical reanalysis data and parametrically generated atmospheric fields. Reanalysis wind and atmospheric pressure data are taken from the ERA5 reanalysis dataset
(Copernicus Climate Change Service (C3S), 2017). The resolution of this data is ~30 km, which may still not adequately resolve the relatively small spatial scales of a hurricane. Therefore, parametric atmospheric wind and pressure fields, generated using the Holland formula (Holland, 1980; Holland et al., 2010) are also used. The Holland formula requires values for the radius of maximum winds, maximum sustained wind speed and central pressure, all of which are obtained from 3-hourly data provided by the International Best Track Archive for Climate Stewardship (IBTrACS) version 4 database (Knapp et al., 2010).
These data were not available for the Ivan case, therefore parametric forcing is only used for Tomas.

Using atmospheric forcing derived using the parametric method, a third hypothetical storm case is considered: a Category 4 hurricane following the same track as Hurricane Tomas, with other parameters taken from a snapshot of the data for Hurricane Ike (Berg, 2014). Specifically, central pressure is set to be a constant 930 mbar, radius of maximum winds to be 28 km and the maximum sustained wind speed to be 68 m s$^{-1}$. It is important to note that a storm of this intensity is rare for the
region, however this experiment is useful for giving an estimate of the storm surge magnitude for a hypothetical extreme hurricane in the future. Here, this case study will be referred to as the 'enhanced Tomas' case.

### 3.3.2 Tide and surge model setup and validation

To model storm surges in the region, the NEMO-surge model is used (Furner et al., 2016); this is a 2-dimensional (i.e. depth-averaged) barotropic configuration of the NEMO model (Madec, 2008). The model domain (Fig. 1) was chosen to include the
whole Caribbean basin and was created by extracting the region from the global ORCA R12 tripolar grid (https://www.geomar.de/en/research/fb1/fb1-od/ocean-models/orca12/). The NEMO-surge configuration used here is a modified version of the NEMO version 3.6 codebase. Ocean physics are represented, for constant density, on a single vertical



level, by removing calls to vertical and tracer processes, as well as using simplified top and bottom stress parameterizations. The output data includes SSH and depth-averaged currents, caused by wind setup and the inverse barometer effect, as well as

tidal forcing. Note that, as the surge model contains no representation of coupled wind wave effects (waves have been modelled independently), related effects such as wave setup and runup are not included.

Tidal forcing was applied at the lateral domain boundaries using 15 harmonic constituents from the TPXO9 dataset (Egbert and Erofeeva, 2002) and the bathymetry has been extracted from the GEBCO dataset (https://www.gebco.net/). The atmospheric forcing used is discussed in Section 3.2 and 3.3. For each case study, two separate model runs were performed: a

run with no atmospheric forcing (tide-only) and a run with both tide and atmospheric forcing (tide + surge). The output from these two runs was then differenced to obtain the non-tidal residuals. Each model run was subject to a spin-up period of three months.

Model validation for the local area around SVG is difficult due to a lack of historical SSH observations. This is especially true during the case study periods (Hurricanes Tomas and Ivan) when there was no recording tide gauge in SVG.

Despite this, testing of the NEMO-surge configuration has shown good results in other regions for both tides and storm surges, for example the Northwest European Shelf Model (Furner et al. 2016).

Data are available at 12 tide gauge sites during 2017, which can be used for validation of model tides. These locations are shown in Fig. 1 and Table SI2. For this purpose, the full year of 2017 was run using the NEMO configuration and the data were analysed to obtain the amplitudes of 4 of the largest harmonic constituents (M2, S2, K1 and O1). Modelled amplitudes

for these four constituents are shown plotted against observed amplitudes at tide gauges in Fig. SI3. From the Great Diurnal Tidal Range (Figure SI4) it can be seen that all tides in the region are small - on the order of (at most) tens of centimetres (microtidal). The modelled values follow the observed values at most locations, representing the magnitude of the four constituents well (Fig. SI3). Other than errors introduced by the model and tide gauges themselves, differences can also be ascribed to errors in the tidal forcing dataset used (TPXO9) and local effects near the tide gauge that the resolution of the

model cannot adequately represent. These comparisons give us confidence in the model's ability to simulate SSH variations on tidal spatial scales.

We can also take a regional look at storm surge activity during 2017. To do this, we have calculated non-tidal residuals (NTR), both modelled and observed, at all tide gauge locations. By estimating errors in NTR, we find an RMSE across all tide gauges of 0.07m. There are also two notable hurricane events (Irma and Maria) in 2017 that are captured by some tide

gauge records in the area. However, these events are far from SVG and significantly more intense than our case study events (and rare in the region around SVG). Fig. 3 shows a comparison of observed and modelled NTR during the passing of Hurricane Maria at the Port-au-Prince tide gauge in Guadeloupe. The model underestimates NTR for this event, possible because to the low resolution of the ERA5 atmospheric forcing being unable to adequately resolve the high winds at the centre of such an intense system. The less intense storms used for our case studies are better resolved by ERA5.





### 3.3.3 Wave model setup and validation

The WAVEWATCH III (WW3) wave model (Tolman, 2009) has been implemented for the Caribbean region with 1/12° resolution (the same domain extent and resolution as the storm surge model, Fig. 1). The WW3 model computes the evolution of the wave spectrum, by solving the wave action equation. The output provides the wave energy at each frequency and propagation direction (30 frequencies and 36 directions were used). Results are typically summarized by the significant wave height (equivalent to the average height of the one-third largest waves), and the (mean and peak) period and direction of waves, obtained as integrated parameters from the wave spectra. Boundary conditions come from the WW3 global model (~0.7°) (described in section 3.2), but forced with ERA5 reanalysis 3-hourly wind fields (0.25°) to better reproduce extreme events.

In the same way as for the storm surge simulations, two historical hurricane events were considered for regional wave modelling, using different methods of wind forcing (see Section 3.3). The regional wave model was initially forced with winds from the ERA5 reanalysis. The modelled significant wave height for Hurricane Tomas (2010) was then validated against the data available from the NOOA National Data Buoy Center (locations shown in Fig.1). Very good correlation was found between simulations and observations with a correlation coefficient about 0.9. The WW3 Caribbean model typically underestimates the significant wave height, (consistent negative bias), but the mean error is of the order of 0.1 m, while the root-mean-square-error is 0.25 – 0.3m (see Table SI4 and Fig. 4). In general, the peaks in wind velocity in the ERA5 dataset, and as a consequence in significant wave height, are underestimated. Therefore, for Hurricane Tomas (2010), we performed an additional simulation using the Holland parametrically generated atmospheric fields (see Section 3.3). However, in the Holland model, background wind fields are not included and this means the wind and waves before and after the storm are under-estimated.

## 4 Results

### 4.1 Sea level changes

Historic observations (Fig. 5) of sea level rise in Caribbean region are limited compared to other regions, which hinders the assessment of coastal impacts and vulnerability in the region (Holgate et al., 2003). For example, tide gauge data from St Vincent are available only for a short period during the years 2013-2016 (www.psmsl.org/cme). The lack of sea level rise data on the coast is especially striking and this is a major barrier to estimating the present rates of sea level change in St Vincent.

We have estimated the basin average rate of sea level rise of $2.8 \pm 0.4$mm yr$^{-1}$ during 1993-2018 using satellite altimetry data sets provided by the Copernicus Marine Environment Monitoring Service (http://marine.copernicus.eu/services-portfolio/access-to-products/?option=com_csw&view=details&product_id=SEALEVEL_GLO_PHY_L4_REP_OBSERVATIONS_008_047). This rate of sea level rise for the region is close to the global number of $3.0 \pm 0.4$mm yr$^{-1}$ (Fig. 6).





Future sea level projections for RCP8.5, including the low probability/high impact scenario (the 95th percentile) are shown on Figure SI4. By 2100 the median sea level rise projections (50th percentile) for SVG with RCP8.5 will be 0.9 m relative to the 1986-2005 reference time period (Church et al., 2013), with up to 2.2m as the 95th percentile, which we have defined as the worst case scenario for sea level rise (Figure SI5).  Estimates for future sea level rise for the Caribbean basin exceed the projections for global sea level rise (Jevrejeva et al., 2016). Sea level rise up to 2.2m would be due to a large contribution from

ice mass loss from both ice sheets, which is very uncertain, but could not be excluded for the risk assessment in coastal area (Jevrejeva et al., 2019; van de Wal et al., 2019). The rate of sea level rise in the region will increase dramatically after 2050 and could be 12mm yr$^{-1}$ (median)  with up to 30mm yr$^{-1}$  (95th percentile)  by 2100 with RCP 8.5, compare to 1.7 -1.9mm yr$^{-1}$ rate of sea level rise in the Caribbean region between 1950-2009 (Torres and Tsimplis, 2013). For the RCP4.5 scenario the median projection of sea level rise in Caribbean basin is 0.6 m, with 0.2-0.9m as 5-95th percentile. Sea level rise with restricted

warming of 1.5 ºC will be 0.5m (median) and up to 0.75m (95th percentile), which is significantly different from the sea level projections with unmitigated warming following emissions scenario RCP8.5 (Jevrejeva et al., 2018; Jackson et al., 2018).

## 4.2  Tides

The Great Diurnal Tidal Range (GDTR) is up to 1m for the Caribbean Sea (Fig. SI4), with a maximum on the continental shelf off Nicaragua, while it does not exceed 0.5m in the Lesser Antilles, except for Trinidad on the continental shelf off Venezuela

in the SE of the region. The GDTR is calculated as the difference in height between mean higher high water (MHHW) and mean lower low water (MLLW), which allows for the variation in tidal range across the Caribbean Sea, due to the strong diurnal inequality and the fact that the semi-diurnal tide has an amphidrome (location of zero tidal amplitude) near Puerto Rico (Greater Antilles).

## 4.3  Waves

### 4.3.1    Wave climatology of the Caribbean Sea

We have simulated the climatological wave conditions of the Caribbean Sea, which are summarised using the results from the 37-year run of the global wave model (Bricheno and Wolf, 2018). Fig. 7 shows the multi-decadal mean of modelled significant wave height (coloured) and mean wave 'from' direction (vectors). The wave climate in the Caribbean is relatively calm, with Hs of the order 1 – 2m on average. With long fetches from the NE, average wave periods (not shown) on the windward

(Atlantic) side can be 6-8 seconds, but are shorter in the lee of islands and within the Caribbean Sea basin. From the wave direction vectors  it can be seen that swell is generated in the North Atlantic, with a prevailing easterly (from) direction, spreading NW into Gulf of Mexico and SW towards the coast of South America. There is some seasonal variability in mean H$_S$, with lowest wave heights observed during October and November, and the largest waves from January – March. However, the mean wave conditions are punctuated by infrequent large storm waves generated by hurricanes. Wave height can reach a





maximum of 12m offshore and 5m at the model cell closest to the coast. During the hurricane season we see the lowest mean

wave climate, but the largest extreme waves.

These rare hurricane events generate large waves (statistical outliers to the generally calm conditions) e.g. from

observations at buoy 42060 (see Table SI3), the mean plus 4 standard deviations is exceeded only 0.07% of the time or 0.26%

of the time at buoy 41044, with waves of $H_S$ greater than 7m. Though hurricanes are mostly limited in area, with largest waves

close to their tracks, they can also generate remote low wave height, long-period swells. Swell waves (caused by tropical or

extra-tropical storms) can be damaging, and affect coasts far from the storm itself (Jury, 2018).

### 4.3.2    Extreme wave events from regional wave model

Fig. 8 shows the maximum significant wave height envelopes for the case study events discussed in Section 3.3.1 for the

regional model. This envelope shows the maximum significant wave height seen at every model cell during the simulated time

period. For the ERA5 forced experiments, significant wave heights are up to 4 m around St Vincent during Hurricane Tomas

(Fig. 8a) and about 6 m at the western side of Grenada during Hurricane Ivan (Fig. 8b). The maximum waves are seen not

along the storm track, but at the right hand side of the track (due to the dynamic fetch effect where wind aligns with the storm

track propagation direction, e.g. Wolf and Woolf, 2006). Thus, even though Ivan was not directly impacting St Vincent, waves

reached 6 m at the western side of the island (Fig. 8b). As already discussed in the surge results section, the spatial and temporal

resolution of the ERA5 atmospheric forcing (30km – hourly) may lead to an underestimation of wind stress, and as a

consequence, of the wave height. Indeed, the experiment with a parametric hurricane Tomas shows an increase in significant

wave height, exceeding 5 m around St Vincent and exceeding 10 m offshore (Fig. 8c). In the event that St Vincent was hit by

a Category 4 hurricane, the significant wave height would exceed 15 m around St Vincent (Fig. 8d).

Fig. 9 shows the significant wave height for Hurricane Tomas at the location of the Argyle International Airport,

located on the east coast of St Vincent. When the WW3 model is forced with ERA5 winds, the significant wave height reaches

4 m, while with the parametric winds it exceeds 5 m. The 'worst case' enhanced wind scenario shows a significant wave height

peak of more than 12 m. In the last two cases, two peaks are clearly visible, showing the passage of the hurricane eye. However,

a model resolution of 12 km means that we are modelling only the offshore wave conditions, whereas some coastal processes,

such as the wave setup and runup, are very important phenomena in the surf-zone. Since they are not represented in the WW3

Caribbean model implementation, this leads to an underestimation of the water levels at the shore.

### 4.4    Storm surge results

Fig. 11 show maximum non-tidal residual envelopes for the case study events discussed in Section 3.3.1. The envelopes show

the maximum non-tidal residual seen at every model cell during the simulated time period. These are useful for quantifying

the storm surge for the whole event period.

For the ERA5 forced experiments (Fig. 10 a, b), non-tidal residuals are small with magnitudes of up to 0.16m around

St Vincent during Hurricane Tomas (panel a) and 0.2m around Grenada during Hurricane Ivan (Fig. 10 b). These are mainly





caused by the inverse barometer effect (see below) which affects SSH only relatively close to the storm track (and centrally along-track), therefore the passing of Ivan had a negligible effect on the SSH around St Vincent. The effects of this storm were mainly seen around Grenada to the south. Fig. 10c shows the results from the Holland-forced Hurricane Tomas run. This figure

shows slightly increased non-tidal residuals around St Vincent (compared to the ERA5 Hurricane Tomas run) of approximately 0.3m. The results above suggest only small increases in SSH around SVG due to the passing of Hurricane Tomas and Ivan. These increases are small when compared to the modelled wave heights and steep terrain of the larger islands. However, they are of the same order of magnitude as the tides (see Section 4.2) and when combined with tide and wave effects, risks may still be exacerbated for coastal populations.

320        Fig.10d shows the maximum non-tidal residual envelope during the 'enhanced Tomas' case study experiment. Non-tidal residuals are significantly higher for this experiment, reaching over 1m at St Vincent. A storm surge of this size, when combined with wind wave effects, poses a significant threat to coastal populations on the islands, especially those that are smaller and more low-lying. Physically, wind setup appears to play a larger part for this stronger tropical cyclone as is evidenced by amplification of the storm surge close to the island itself.

325        Small non-tidal residuals seen in the model output have a number of possible explanations. It can be seen in Fig. 1 that the local bathymetry is steep and deep near the islands. The lack of a continental shelf means that there is little shallow water fetch and, as wind stress is inversely proportional to the ocean depth, wind setup is small. Other areas that experience similar storms but significantly higher surges, e.g. the south coast of the US (Berg, 2014; Beven and Kimberlain, 2009), have much wider areas of shallow water/continental shelf. The small spatial extent of the islands may also mean that wind setup is

reduced due to ocean currents diverging and going around the islands.

       Little amplification in SSH is seen very close to the islands, suggesting that much of the non-tidal residuals are generated by the inverse barometer effect. For example, the inverse barometer effect can be estimated using the basic relationship of 1cm of SSH for every 1mbar of pressure difference. This is a simple estimation but would result in an increase in SSH of around 0.83m at the storm centre for the 'enhanced Tomas' case. This would account for most of the storm surge

seen in the model results for this experiment. The constraints of the model and forcing data must also be considered. A model resolution of 12km may mean that the narrow shelf (less than a few km) around the islands is not being adequately represented, leading to an underestimation of wind stress, currents and consequently non-tidal residuals.

## 5      Combined vulnerability index

Coastal threats from the sea can include flooding and coastal erosion caused by steadily rising sea levels as well as changes on

other time-scales e.g. changing tides, storm surges and waves. The challenge of combining these different processes into a single vulnerability index, which can allow us to compare different types of coastline on a national, regional and global scale, is addressed here.





Coastal vulnerability may be defined as the degree to which a system is susceptible to, or unable to cope with, adverse effects of climate change, including climate variability and extremes. In IPCC (2012), vulnerability is defined as the propensity or
predisposition to be adversely affected.

An accurate and quantitative approach to predicting coastal change is difficult to establish. Even the kinds of data necessary to predict shoreline response are the subject of scientific debate. A number of predictive approaches have been proposed (National Research Council, 1990 and 1995), including:

- extrapolation of historical data (e.g., coastal erosion rates),
- static inundation modelling,
- application of a simple geometric model (e.g., the Bruun Rule),
- application of a sediment dynamics/budget model, or
- Monte Carlo (probabilistic) simulation based on parameterized physical forcing variables.

However, each of these approaches has inadequacies or can be invalid for certain applications (National Research Council,
1990). Additionally, shoreline response to sea-level change is further complicated by human modification of the natural coast such as beach nourishment projects, and engineered structures such as seawalls, revetments, groins, and jetties. Understanding how a natural or modified coast will respond to sea-level change is essential to preserving vulnerable coastal resources.

Here we adapt the Coastal Vulnerability Index (CVI), derived by Thieler and Hammar-Klose (1999, hereafter THK), by dividing it into the geological variables (geomorphology, shoreline erosion/accretion rate and coastal slope) and the physical
variables (relative SLR rate, mean significant wave height and mean tidal range). The geological variables require specifc knowledge of the coastal typology (which can be calculated for detailed coastal studies on a local, national or regional scale), whereas the physical variables can be seen rather as external drivers and can be quantified from the regional models described above.

THK define the CVI as a combination of geological and physical variables as follows (with a slight re-definition of
parameters:

$$CVI = \sqrt{\frac{(a*b*c)*(d*e*f)}{6}}$$

where,
a = geomorphology factor
b = coastal slope factor
c = shoreline erosion/accretion rate
d = relative sea-level rise rate factor
e = mean tide range factor





375          f = mean wave height factor.

Here a,b,c are geological parameters: and d,e,f are external physical parameters.

        Table 1 shows the six variables described by THK, which include both quantitative and qualitative information. The

five quantitative variables are assigned a vulnerability ranking based on their actual values, whereas the non-numerical

geomorphology variable is ranked qualitatively according to the relative resistance of a given landform to erosion. Shorelines

with erosion/accretion rates between -1.0 and +1.0m yr$^{-1}$ are ranked as being of moderate vulnerability in terms of that

particular variable. Increasingly higher erosion or accretion rates are ranked as correspondingly higher or lower vulnerability.

Regional coastal slopes range from very high vulnerability, <4.59 percent, to very low vulnerability at values >14.7 percent.

The rate of relative sea-level change is ranked using the modern rate of eustatic rise (1.8mm yr$^{-1}$) as very low vulnerability.

Since this is a global or "background" rate common to all shorelines, the sea-level rise ranking reflects primarily local to

regional isostatic or tectonic adjustment. Mean wave height contributions to vulnerability range from very low (<1.1 m) to

very high (>2.6 m). Tidal range is ranked such that micro-tidal (<1m tidal range) coasts are very high vulnerability and macro-

tidal (>6m tidal range) coasts are very low vulnerability.

        There have been various developments of the indicator-based approach e.g. Ramieri et al (2011). New models have

been developed for the assessment of coastal vulnerability at the global to national level e.g. the DIVA model (Hinkel and

Klein, 2009).

        Following Özyurt and Ergin (2010, hereafter OE2010) an additive model has been used to combine parameters, with

normalisation, which allows more quantitative inter-comparison between different coastline, with respect to exposure and

resilience. OE2010 added further physical parameters, representing hydrological parameters, such as groundwater and river

discharge, and also add a set of human parameters, representing anthropogenic changes, such as hard defences, reduction in

sediment supply and land use. This divides the model into 2 subsets:

        CVI = (½ CVIPP + ½CVIHP)/CVILV, where CVILV is the least vulnerable location, used as a normalisation factor.

Here CVIPP is derived from the physical parameters and CVIHP from the human parameters, such that

CVIPP = $\sum PP_n.R_n$ (summed over all n physical factors considered, $PP_n$, with weighting factors $R_n$) and CVIHP = $\sum HP_m.R_m$

(summed over all m human factors considered, $HP_m$, with weighting factors $R_m$).

        If we ignore the human interventions at the coast and the coastal geomorphological factors (which we have not derived

at the regional scale), we can calculate an external physical exposure factor, including the rate of sea level rise, the wave

climate and tidal range, which we here refer to as a Combined Vulnerability Index (CBVI). We have calculated the

vulnerability indices, for each factor, for all coastal points around the Caribbean, using the data derived in sections 3 and 4 and

the parameter ranges in Table 1. As the tide is microtidal, the tidal vulnerability index is a maximum nearly everywhere.

Likewise, there is a saturation of the index for the rate of sea level rise, because the present-day rate of 2.8 mm yr$^{-1}$ exceeds





the maximum in the original definition everywhere and future projections lead to much higher rates. There is some variation

of the wave index, showing larger vulnerability at more exposed coasts. We have also chosen to include an additional storminess index, to represent the amount of exposure to severe storms: the data used for this is a 30 year climatology from the ERA5 reanalysis dataset. From this, we have chosen the mean annual maximum wind-speed (WSMAM) as the parameter which best illustrates the variation in storminess across the region. The range (1-5) has been set as follows:  WSMAM) <10m s$^{-1}$, index = 1; 10< WSMAM <15, index = 2; 15< WSMAM <20, index = 3; 20< WSMAM <25, index = 4;  WSMAM >25m

s$^{-1}$, index = 5. The more southerly coastlines, close to South America, have the lowest vulnerability to storm winds, then the vulnerability increases northward along the Windward and Leeward Islands, reaching a maximum around Anguilla. The Gulf of Mexico shows generally higher vulnerability and the Turks and Caicos Islands (outside the Caribbean Island arc), are particularly susceptible to strong winds, being along the main tropical storm track. See Fig. 11a for the variation of the CBVI without winds, over the Caribbean, while Fig. 11b shows the CBVI including winds. Note that, for the present, a uniform

weighting factor (i.e. 1) is applied to all the 4 indices, in the absence of a better way of assessing what this should be.

## 6       Discussion

In this study we have quantified individual processes contributing to coastal hazards in the Caribbean, suggesting that in the present-day climate the greatest hazards at the coast are waves, with observed and modelled height for extreme events up to 5 m near the coast and maximum of 12 m offshore. Note that the modelled waves are expected to be underestimated during the

most extreme events, since winds are under-estimated in the atmospheric model, due to limited spatial resolution, particularly for the most extreme events. For the synoptic view, a reanalysis data set covers the full range of conditions across the Caribbean Sea. Due to the coarseness of the model grid, some small islands are unresolved, however the missing land is represented in the model by a partial obstruction (Tolman, 2009). A sheltering effect can be seen around the Windward Islands (62° W, 10 - 16° N), even though the archipelago is not explicitly present as 'land points'.

430        We have simulated storm surges for the past extreme events associated with Hurricane Ivan (2004) and Hurricane Tomas (2010), providing estimates of storm surge up to 0.16m in the coastal areas of SVG. An additional scenario,  hurricane that represents a category 4 event on a direct trajectory for SVG, we simulated storm surge which might reach up to 1.5m for a direct hurricane impact depending on the underwater topography. Tidal range for most of locations is less than 0.5m, due to deep water close to shore. The tide can be amplified in shallow continental shelves e.g. east of Nicaragua and around Trinidad.

435        The regional sea level trend calculated from satellite altimetry data for the period 1993-2018 is 2.8± 0.4mm yr$^{-1}$ compared to the global mean trend of 3.0± 0.4mm yr$^{-1}$, which is in a good agreement from previous studies with shorter time period (Torres and Tsimplis, 2013; Palanisamy et al., 2012). Several estimates from tide gauge records provide a wide range (0.3-12mm yr$^{-1}$) of past sea level rise rates for individual locations with different observational time period (Torres and Tsimplis, 2013; Palanisamy et al., 2012). The large difference in rate for individual locations is explained by the lack of





consistent observations, gaps in time series, lack of information about the vertical land movement in location of tide gauge (Holgate et al., 2003; Palanisamy et al., 2012; Torres et al., 2013).

The main limitations to estimating present day processes contributing to coastal hazards are:

- Lack of the observational data for assessment of current changes in sea level, storm surges and waves. In addition, lack of observations is limiting model calibration. It seems there is a relation between model performance and the
number of gaps in the observations. The best fits are for continuous records, the ones with big gaps do not do so well.

- Lack of sufficient forcing (e.g. local wind) to simulate extreme events at the coast is a challenge to reproduce past events.

- We present a vulnerability metric for the region, allowing intercomparison of the risks from marine hazards in different locations. However, we notice that approach needs to be updated taking into account the larger changes in
physical variables (e.g. relative SLR rate) for different climate scenarios.

- In this study we have quantified individual processes contributing to coastal hazards on the regional scale, for the Caribbean. We have not investigated the interaction between these processes, e.g. between water level, waves and storm surges. In general these interaction effects are secondary, at a regional scale, but may be more significant at a local scale. For steep volcanic islands, large waves are often the dominant contributor to increased water levels during
storms, as deep water close to shore permits large waves to reach the coast while limiting the generation of storm surges.

- Some issues are not addressed here, such as coastal inundation, including wave runup, which requires detailed knowledge of the local coastal topography. Some preliminary work on this has been carried out for St Vincent, using reconstructed beach profiles (Prime et al., 2019) and this will be addressed in future work.

The Combined Vulnerability Index calculated on the base of current climate conditions demonstrates that islands on the eastern part of the basin are more vulnerable, due to impact from wind and waves. However, with an increasing rate of sea level rise by 2100 for all emission scenarios, sea level rise is imposing a large hazard in the region. Extreme sea levels associated with hurricanes, tropical cyclones and waves are projected to change in a warming climate. Nevertheless, recent results from Morim et al. (2019) for wave projections under future climate scenarios for the eastern Caribbean Sea (for the more extreme 'business-
as-usual' RCP8.5 emissions scenario) shows no change or a slight reduction (<5%) in mean and extreme wave height. When divided into seasons there can be seen a small increase (<5%) in mean wave height during the height of the summer (JJA), which also coincides with the hurricane season.

In our study we have demonstrated that a storm surge might reach 1.5m for a direct hurricane impact depending on the underwater topography under an additional scenario hurricane that represents a category 4 event. Projections for the 21st
century indicate that it is likely that the global frequency of tropical cyclones will either decrease or remain essentially unchanged, along with a likely increase in both global mean tropical cyclone maximum wind speed and rain rates (IPCC, 2013). The frequency of the most intense storms will more likely than not increase in some basins. More extreme precipitation is projected near the centres of tropical cyclones making landfall in North and Central America (IPCC, 2013).




A comparative study of the impact of sea level rise on coastal inundation across 84 developing countries showed that the
greatest vulnerability to a 1m sea level rise in terms of inundation of land area was located in East Asia and the Pacific,
followed by South Asia, Latin America, and the Caribbean, the Middle East and North Africa, and finally sub-Saharan Africa
(Dasgupta et al., 2009). Future sea level rise will pose a significant threat to the coastal infrastructure, settlements, beaches,
ecosystems and economic activity in coastal areas of SIDs. In St Vincent sea level rise of 1m will place 10% of the main tourist
infrastructure at risk, along with 67% of sea port infrastructure (CARIBSAVE, 2012).  In addition, sea level rise will alter the
erosion of the coastline, with approximately 100 m of erosion resulted from 1 m of sea level rise CARIBSAVE, 2012). Up to
70% of the major tourist resorts will be impacted by the disappearance of the beaches due to sea level rise and erosion
(CARIBSAVE, 2012).

## 7 Conclusion

For Caribbean island nations it is crucial that the spatial variability in hazardous wave and water level conditions and the
contributions of different processes in causing hazardous conditions are understood, in order to prepare and resource island-
wide coastal climate resilience plans. In our study, we have explored the role of waves, storm surges and sea level rise for the
Caribbean region with a focus on coastal impacts in the eastern Caribbean, using St Vincent and the Grenadines (SVG) as an
example. We identified that currently the greatest hazard is waves, with up to 12 m wave height offshore and 5 m near the
coast for extreme events (e.g. Hurricane Ivan/Tomas). For the same extreme events the storm surge height may be up to 1.5m.
Sea level rise since 1993 is about 0.1m, which is close to the global sea level rise estimate for the same time period. However,
sea level rise is accelerating and by 2100 projected sea level rise could exceed 2m by 2100 with RCP8.5 scenario.

Coastal communities in the Caribbean are facing unprecedented challenges due to climate change. There is a need for
commitment to sustained, systematic and complementary coastal and near shore measurements of waves and sea levels to
improve the skill of forecasting (including early warning) systems, and future projections by the international scientific
community, international and local government bodies. Regular stakeholder workshops to gather evidence on local needs and
issues are an important part of the work, and have helped to guide the direction of research and design of practical outputs.
Close contact between the governments of the SIDS and international/local research community is crucial to provide updated
scientific evidence for decisions associated with shoreline management to build climate resilient communities and
infrastructure. Our study provides scientific evidence for policy makers, scientists, and local communities to actively prepare
for and protect against climate change.

**Acknowledgement**

This work was supported by the UK government through the Commonwealth Marine Economies Programme, which aims to
enable safe and sustainable marine economies across Commonwealth Small Island Developing States.





The following organizations provided best track data which are included in the IBTrACS data set: Australian Bureau of Meteorology; Chinese Meteorological Agency; Shanghai Typhoon Institute; Hong Kong Observatory; Joint Typhoon Warning Center; RSMC Honolulu, HI, USA (NOAA Central Pacific Hurricane Center); RSMC La Reunion; RSMC Miami, FL, USA (NOAA Tropical Prediction Center) (HURDAT); RSMC Nadi, Fiji; RSMC New Delhi, India; RSMC Tokyo, Japan; TCWC Wellington, New Zealand. Special thanks go to the NCDC Climate Database Modernization Program (CDMP) for digitizing hardcopies of best track data and Charlie Neumann, for making his Southern Hemisphere dataset available for this work. Ocean Heat Content data for 1988-2017 (defined as an anomaly) were taken from https://www.nodc.noaa.gov/OC5/3M_HEAT_CONTENT/, accessed on 27 February 2019. We acknowledge the use of UK national supercomputing resource ARCHER which was used to perform the wave and surge model runs.

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



Table 1: Ranges for Vulnerability Ranking of Variables on the Atlantic/Caribbean Coast (from THK)

| Variables | Very Low 1 | Low 2 | Moderate 3 | High 4 | Very High 5 |
|---|---|---|---|---|---|
| Geomorphology | Rocky cliffed coasts, Fjords | Medium cliffs, Indented coasts | Low cliffs, Glacial drift, Alluvial plains | Cobble Beaches, Estuary, Lagoon | Barrier beaches, Sand beaches, Salt marsh, Mud flats, Deltas, Mangrove, Coral reefs |
| Shoreline erosion/accretion (m yr$^{-1}$) | > 2.0 | 1.0 - 2.0 | -1.0 - 1.0 | -2.0 - -1.0 | < -2.0 |
| Coastal slope (%) | > 14.70 | 10.90 - 14.69 | 7.75 - 10.89 | 4.60 - 7.74 | < 4.59 |
| Rate of sea level changes (mm yr$^{-1}$) | < 1.8 | 1.8 - 2.5 | 2.5 - 3.0 | 3.0 - 3.4 | > 3.4 |
| Mean wave height (m) | < 1.1 | 1.1 - 2.0 | 2.0 - 2.25 | 2.25 - 2.60 | > 2.6 |
| Mean tide range (m) | > 6.0 | 4.0 - 6.0 | 2.0 - 4.0 | 1.0 - 2.0 | < 1.0 |

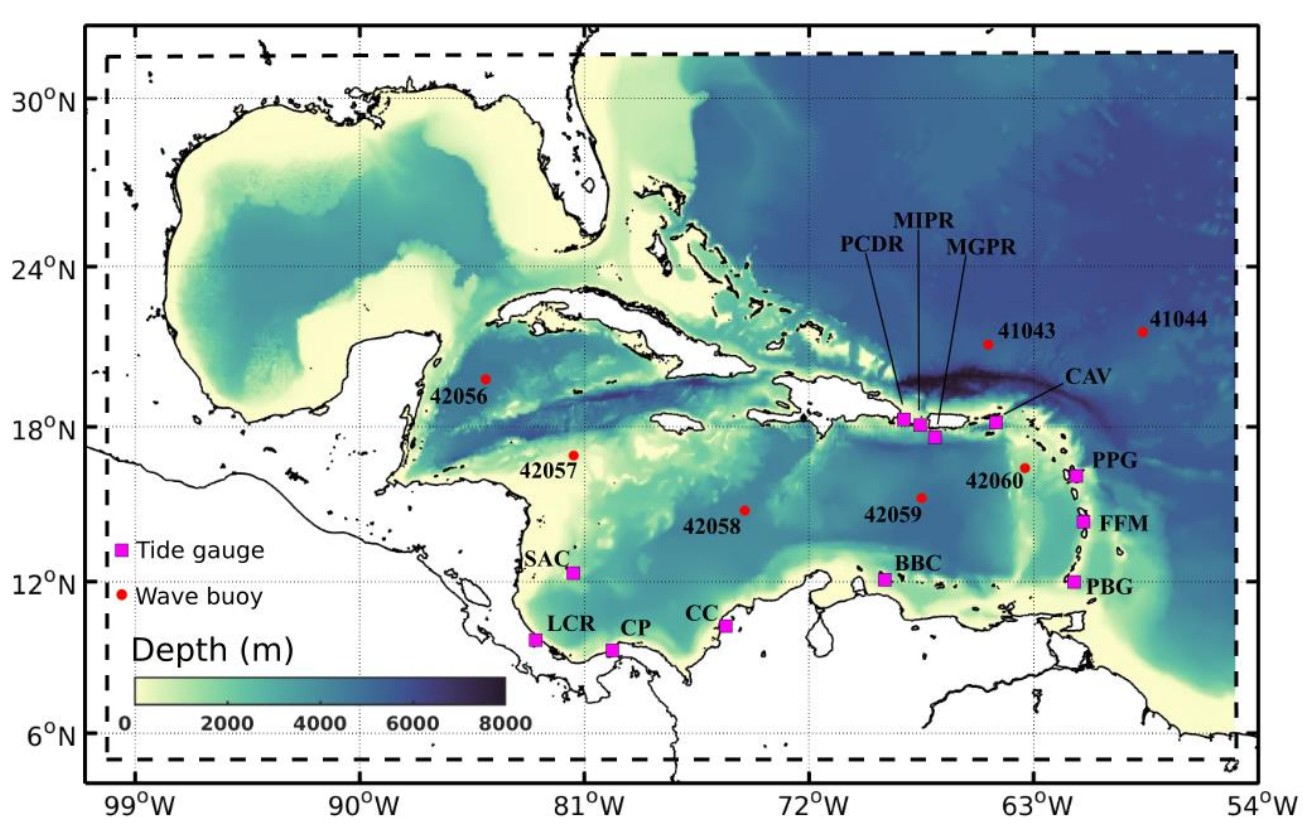

**Figure 1: Map of the Caribbean Sea, showing bathymetry and locations of 12 tide gauges and 7 wave buoys used for models validation. Tables SI2 and SI3 give the locations of the tide gauges and wave buoys respectively.**





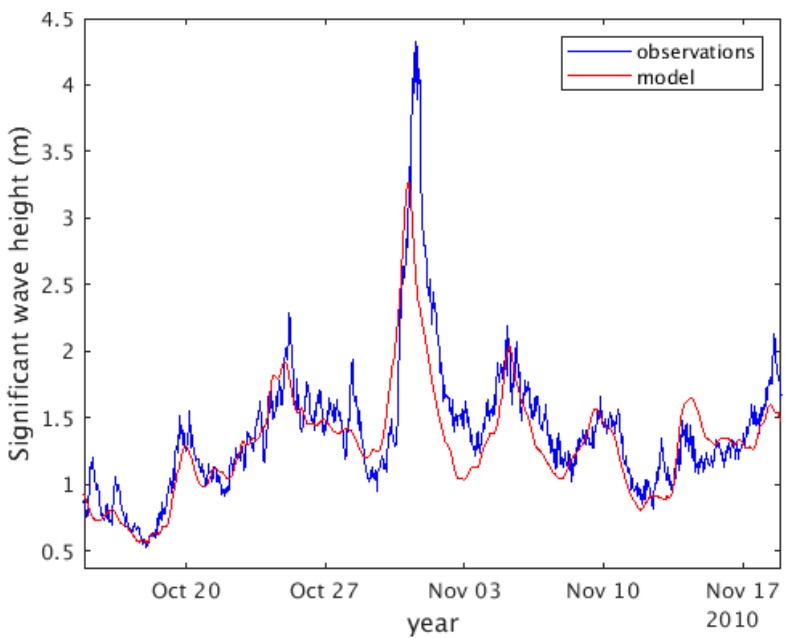

**Figure 2: Time series showing a zoom to the dates of Hurricane Tomas (Oct 2010) from the global model and observations at the buoy 42060 (see Fig. 1 for location).**

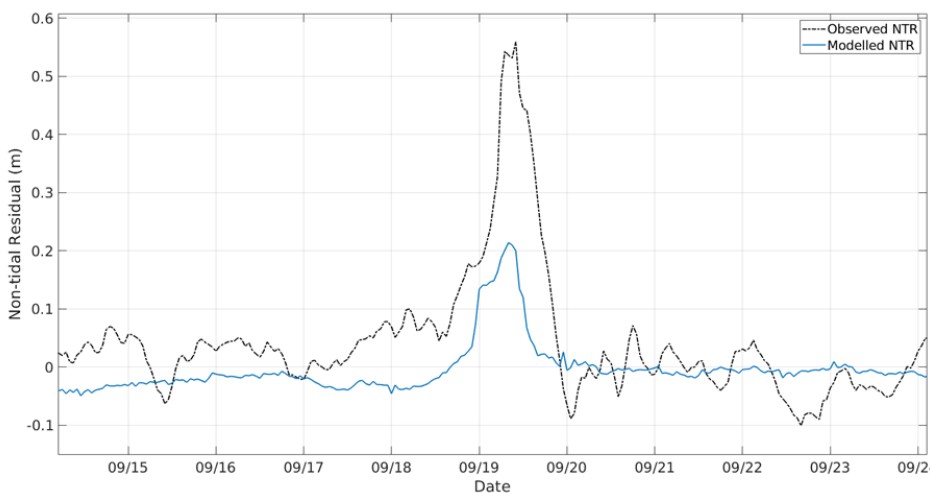


**Figure 3: Modelled and observed non-tidal residuals (m) during the passing of Hurricane Maria (2017) over the Port-au-Prince tide gauge (Port-au-Prince, Guadeloupe).**



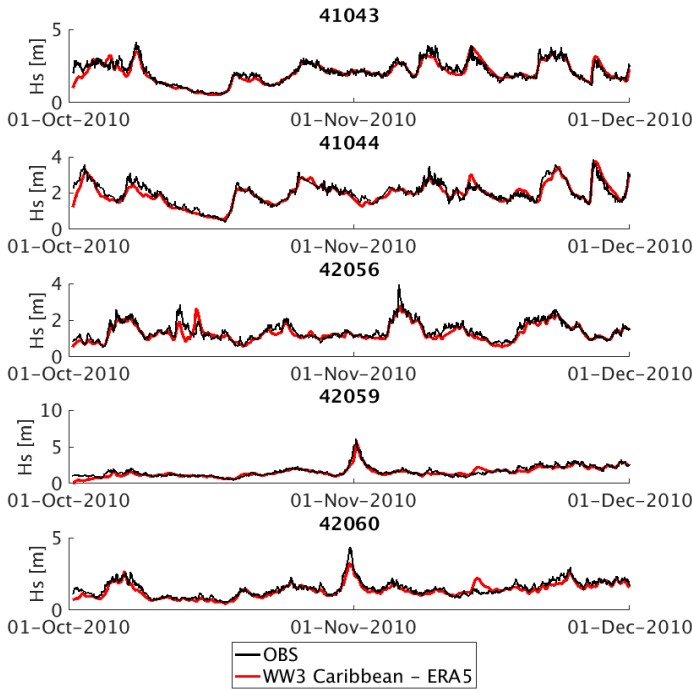

**Figure 4: Comparison between the significant wave heights observed by 5 buoys from the NOOA National Data Buoy Center and modelled by the WW3 Caribbean model forced by ERA5 reanalysis winds. Validation is for October 2010 and November 2010, covering the passage of Hurricane Tomas (29 October 2010 – 14 November 2010).**

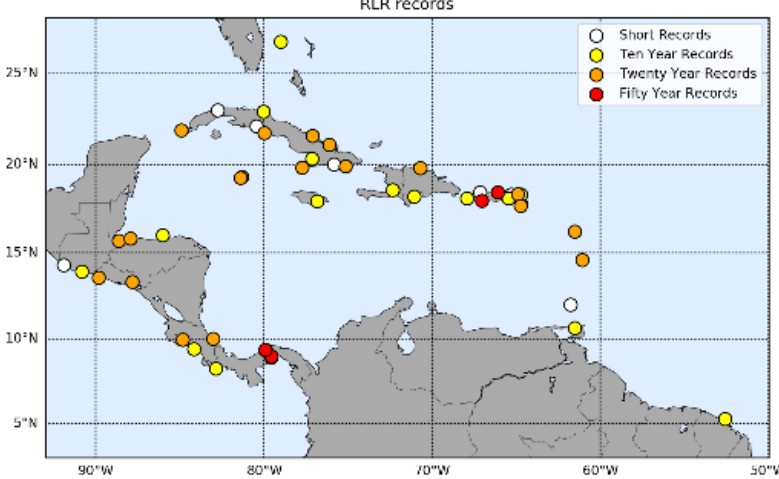


**Figure 5: Locations of tide gauge records (historical monthly mean sea level data with revised local reference (RLR)) available from the Permanent Service for Mean Sea Level (PSMSL).**

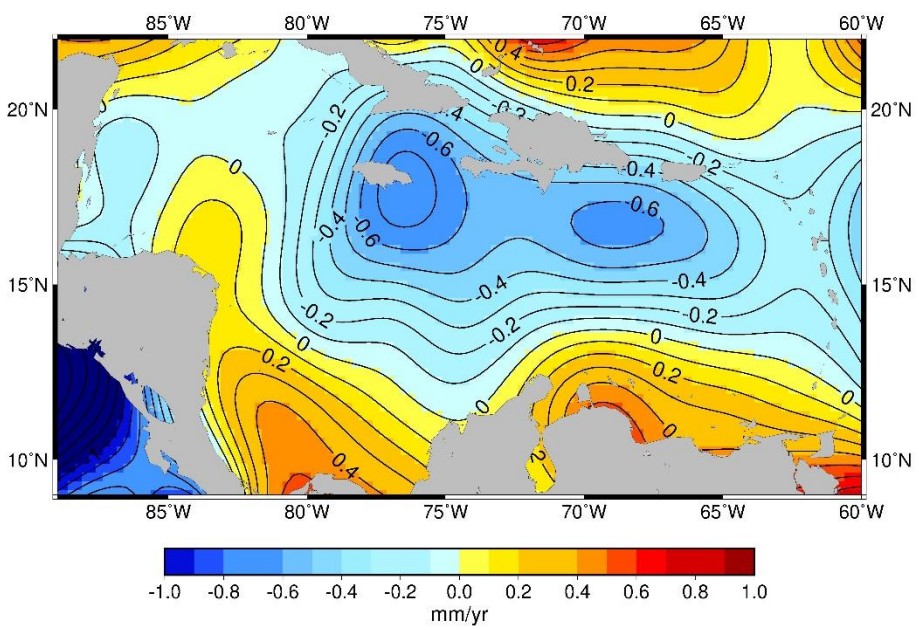

**Figure 6: Sea level trends in the Caribbean basin (whereas the global mean trend is 3.00 +/- 0.4mm yr$^{-1}$ is removed), calculated using satellite altimetry data 1993-2018.**


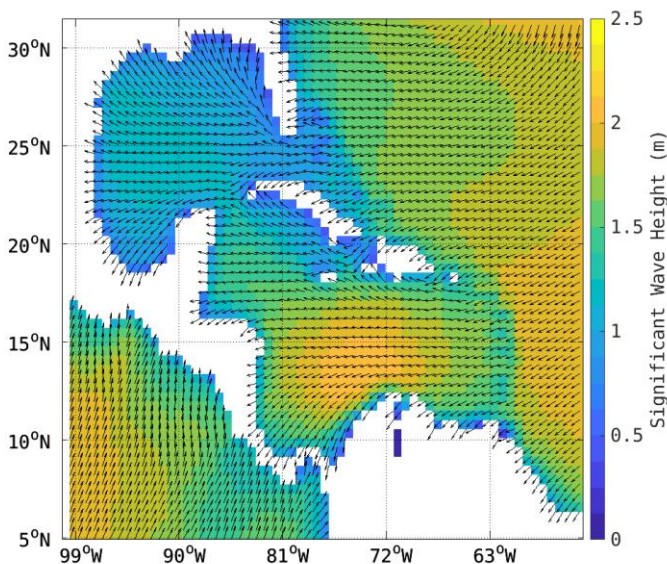

**Figure 7: Mean significant wave height (m, colour shading) and mean direction (vectors) from the global 37-year**
**historic climatology wave model run. Locations of wave buoys used for model validation are marked as filled red circles.**




**Figure 8: Maximum significant wave height simulated by the WW3 Caribbean model during a) Hurricane Ivan (2004) and b) Hurricane Tomas (2010) forced by ERA5 and c) forced by the Holland model parametric wind field and d) forced by the enhanced wind field.**





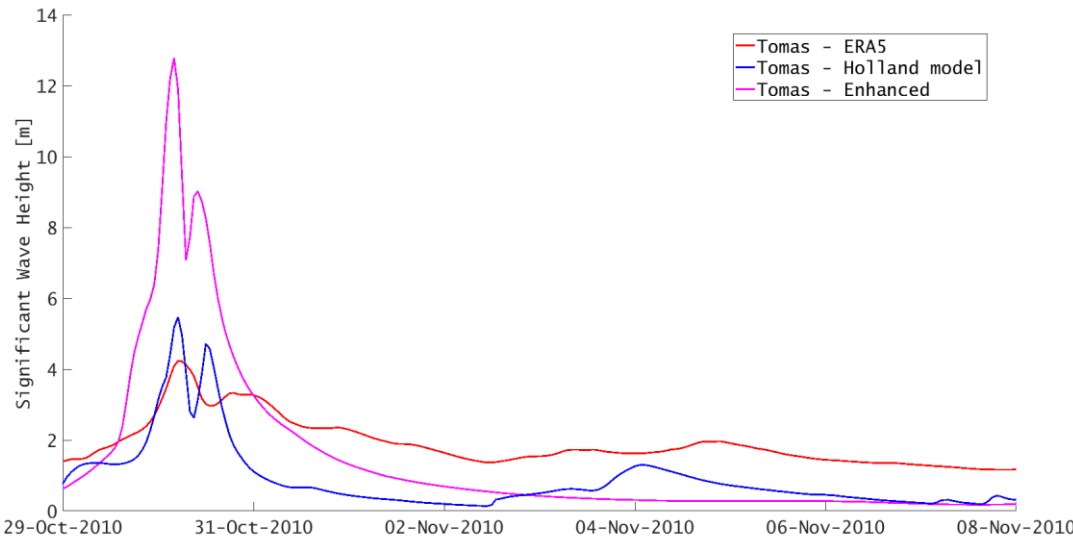

**Figure 9: Significant wave height at Argyle airport during Hurricane Tomas: red line – wave model run forced with**
**the ERA5 reanalysis; blue line – wave model forced with the Holland's model parametric winds; pink line – wave model**
**forced with "worst case scenario" enhanced winds.**


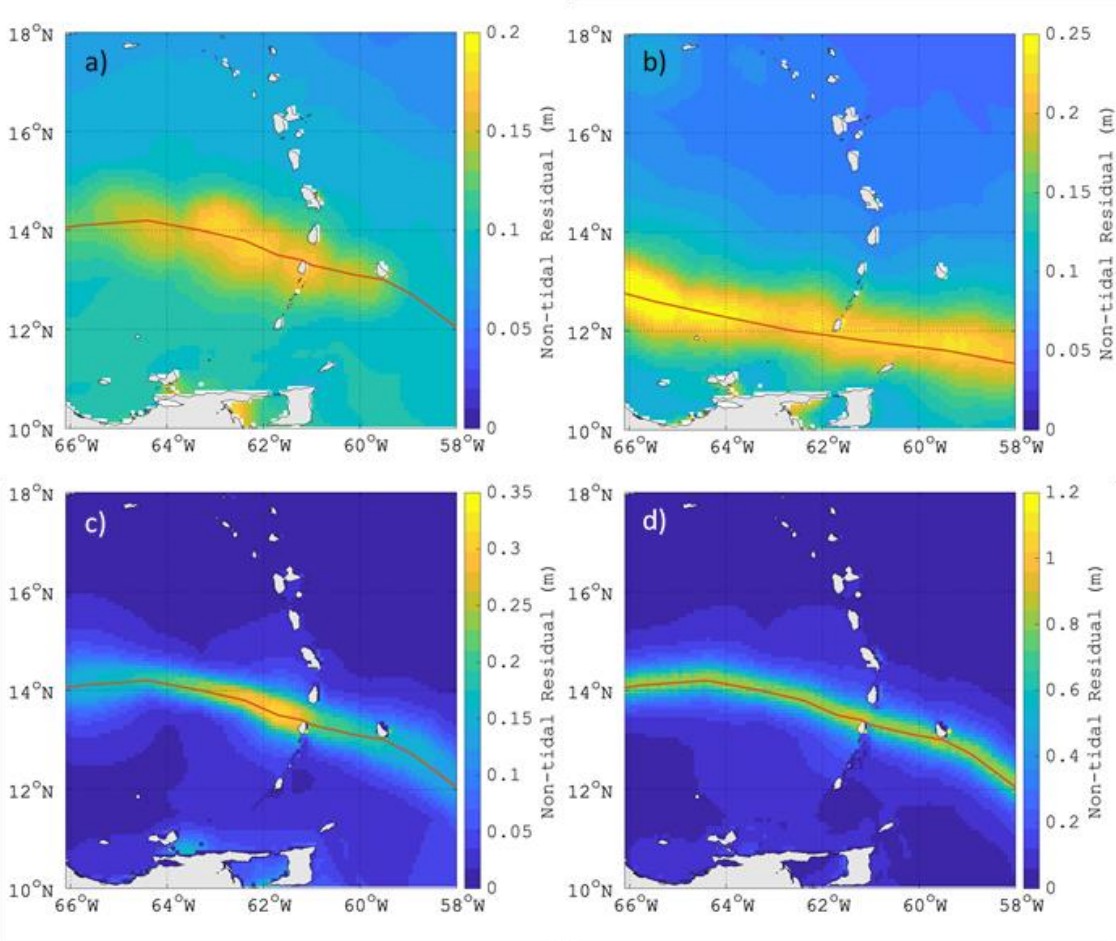

**Figure 10: Maximum non-tidal residual envelopes for four model runs: a) ERA5-forced Hurricane Tomas, b) ERA5-forced Hurricane Ivan, c) Holland-forced Hurricane Tomas and d) 'Enhanced Tomas' case study.**



**Figure 11: Coastal vulnerability metrics; a) combined based on mean annual rate of sea level rise, mean annual maximum waves and Great Diurnal Tidal Range and b) as a) plus maximum winds.**