# Peer review of "Quantifying processes contributing to coastal hazards to inform coastal climate resilience assessments, demonstrated for the Caribbean Sea"

_Natural Hazards and Earth System Sciences, 2020_

## Referee Comment (RC1) · Anonymous Referee #1 · 9 Apr 2020

Jevrejeva et al., assess coastal impact of waves, storm surges and sea level rise in the Caribbean region, with a focus on coastal impacts in the eastern Caribbean islands (St Vincent and the Grenadines). The proposed methodology can aid in decision-making about coastal adaptation strategies, especially in these small island developing states. This work addresses relevant scientific questions and of interest to the NHESS journal's audience. Although I'm positive, a few issues should be addressed before accepting this work, which I have summarized below.

The main challenge to estimate the vulnerability of the Lesser Antilles is linked to the very contrasting morphologies of these small islands, which influence the submersion

[Figure]

dynamics significantly along the coastline (see for example Duvat et al. 2019). An accurate bathymetry is a key factor in this region. The authors used GEBCO, which has a resolution of the order of the kilometer and a very low precision in shallow water. In this context, could the authors explain what is the accuracy/representativeness of their model's outputs and their vulnerability index estimate?

Duvat, V., Pillet, V., Volto, N., Krien, Y., Cécé, R., & Bernard, D. (2019). High human influence on beach response to tropical cyclones in small islands: Saint-Martin Island, Lesser Antilles. Geomorphology, 325, 70-91.

In the same way, how can the authors be sure that the changes in the non-tidal residuals (section 4.4) during Hurricane Tomas and Ivan were mainly caused by the inverse barometer effect and not due to the model's low resolution (wind+bathymetry)?

Comparisons with measurements are too qualitative. The buoys used for validation seem to be very far from the cyclone tracks. Could the authors please comment on this?

Line 175: the radius of maximum winds for Ivan could be estimated from the HURDAT Re-analysis dataset (https://www.aoml.noaa.gov/hrd/hurdat/Data_Storm.html)

How are the inputs from rivers and rainfall incorporated into the model grid? (boundary conditions?). What about other factors that contribute to total water levels such as vertical land movement (tectonic/seismic activity and anthropic) and wind waves (setup and runup) ? Could the authors provide more details/discussions on that? What are the consequences on their results of not considering these important processes?

The literature review seems poor, I suggest to enlarge it looking at the recent advances in this region. In addition, there is no scientific comparison/discussion in respect of recent storm surge research.

Among others:

Zahibo, N., Pelinovsky, E., Talipova, T., Rabinovich, A., Kurkin, A., & Nikolkina, I.

(2007). Statistical analysis of cyclone hazard for Guadeloupe, Lesser Antilles. Atmospheric research, 84(1), 13-29.

Krien, Y., Dudon, B., Roger, J., & Zahibo, N. (2015). Probabilistic hurricane-induced storm surge hazard assessment in Guadeloupe, Lesser Antilles. Natural Hazards & Earth System Sciences Discussions, 3(1).

Krien, Y., Dudon, B., Roger, J., Arnaud, G., & Zahibo, N. (2017). Assessing storm surge hazard and impact of sea level rise in the Lesser Antilles case study of Martinique. Natural Hazards & Earth System Sciences, 17(9).

Kennedy, A. B., Gravois, U., Zachry, B. C., Westerink, J. J., Hope, M. E., Dietrich, J. C., ... & Dean, R. G. (2011). Origin of the Hurricane Ike forerunner surge. Geophysical Research Letters, 38(8).

The vulnerability definition that is used by the authors remains vague. Please, elaborate more on the concepts of risk, exposure, vulnerability... In the last section, a Coastal Vulnerability Index (CVI), based on the methodology used by Thieler and Hammar-Klose (1999) is presented, gathering six variables : geomorphology; coastal slope; shorline erosion/accretion/ relative slr; mean tide and mean wave. However, geomorphology is not considered in the study, slr is constant for all the study region (1.8mm/yr), no justification/references is given for the choice of the shoreline changes (between -1.0 and +1.0m yr-1 ), so only mean weight height and tidal range provide relevant information to compute the CVI. The same observation is made concerning the choice of the CVIPP from OE210, where the human interventions at the coast and the coastal geomorphological factors are ignored. At the end, what is the meaning of this vulnerability index (figure 11)?

Minor comments:

The figures 1 and 11 are not well constructed neither clear for understanding. For example, in Figure 1 : the names of countries and the names tide gauge stations are

missing...

L217 : please correct : Is the comparison made with Pointe-à-Pitre tide gauge located in Guadeloupe or with the Port au Prince tide gauge located in Haiti ?

---

## Referee Comment (RC2) · Anonymous Referee #2 · 19 Apr 2020

The manuscript details a methodology for regional assessment of coastal impacts due to waves, storm surges and sea level rise. The study focuses on Caribbean SIDS (Small Island Developing States), which are particularly vulnerable to coastal climate change, since they strongly rely on the preservation of the coastal zone and they are also prone to natural disasters. The adopted approach can provide information for the design and implementation of the requisite coastal adaptation strategies. I recommend the publication of the manuscript following some clarifications and minor corrections.

The terms risk, exposure and vulnerability are used in a confusing way (e.g. Pg. 1 lines 18-19: "We introduce a Combined Vulnerability Index, which allows a quantitative

assessment of relative risk across the region, showing that sea level rise is the most important risk factor everywhere" and Pg. 13 lines 404-405: "we can calculate an external physical exposure factor, including the rate of sea level rise, the wave climate and tidal range, which we here refer to as a Combined Vulnerability Index (CBVI)"). Please clarify better these concepts.

An analysis was made regarding hurricane induced storm surges and a Combined Vulnerability Index (CBVI) is proposed for marine hazards; however storm surges are not included on the proposed CBVI. Could the authors comment on that?

The names of the countries and the locations mentioned in section 2 should be added in Figure 1.

Figure 5 does not include very important information for the manuscript; it could be transferred in the supplementary material or it could be combined with figure 6.

The caption in Figure 7 (Pg.27, line 686) mentions "Locations of wave buoys used for model validation are marked as filled red circles" but the locations are missing from the figure.

The map in Figure 7 looks deformed, the scale of the figure should be corrected.

Caption of figure 8 (Pg. 28, lines 696-697): It is not clear that Fig. 8c and 8d concern Hurricane Tomas.

Figure 11 is not very clear, needs to be improved.

Figure SI1 and Table SI1 of the supplementary material is not mentioned in the manuscript.

Pg. 9, lines 250-251: "Future sea level projections for RCP8.5, including the low probability/high impact scenario (the 95th percentile) are shown on Figure SI4". It's Figure SI5 not Figure SI4.

Pg. 10, line 307: "Fig. 11 show maximum non-tidal residual envelopes for the case

study events discussed in Section 3.3.1". It's Fig. 10 that depicts the non-tidal residual envelopes, not Fig. 11.

Caption of figure 6 (Pg. 26, line 680): Please correct the typing error "(whereas the global mean trend is 3.00+/-0.4mm yr-1 is removed)"

―――――――――――――――

---

## Author Response (AR1)

Quantifying processes contributing to coastal hazards to inform marine climate resilience assessments, demonstrated for the Caribbean Sea

MS No.: nhess-2020-46

Reply to Reviewer 1 comments

Anonymous Referee #1

Jevrejeva et al., assess coastal impact of waves, storm surges and sea level rise in the Caribbean region, with a focus on coastal impacts in the eastern Caribbean islands (St Vincent and the Grenadines). The proposed methodology can aid in decision-making about coastal adaptation strategies, especially in these small island developing states. This work addresses relevant scientific questions and of interest to the NHESS journal's audience. Although I'm positive, a few issues should be addressed before accepting this work, which I have summarized below.

1. The main challenge to estimate the vulnerability of the Lesser Antilles is linked to the very contrasting morphologies of these small islands, which influence the submersion dynamics significantly along the coastline (see for example Duvat et al. 2019). An accurate bathymetry is a key factor in this region. The authors used GEBCO, which has a resolution of the order of the kilometer and a very low precision in shallow water. In this context, could the authors explain what is the accuracy/representativeness of their model's outputs and their vulnerability index estimate?

Reference: Duvat, V., Pillet, V., Volto, N., Krien, Y., Cécé, R., & Bernard, D. (2019). High human influence on beach response to tropical cyclones in small islands: Saint-Martin Island, Lesser Antilles. Geomorphology, 325, 70-91.

*Reply:*

*Thanks for pointing out this paper, which explores detailed coastal impacts for the Lesser Antilles French/Dutch island of Saint-Martin/Sint Maarten at the northern end of the Antilles arc (at 18N). While we fully accept that the coastal impacts on each island are governed by their coastal morphology (and intend to explore this in a second paper, referring in more detail to St Vincent); in the present paper we are focussing on the exposure of the whole Caribbean Sea to marine hazards rather than trying to examine the detailed impacts for individual islands (although we do refer to the case study for SVG to illustrate local impacts as a consequence of regional processes). This does present a challenge, particularly for estimating coastal storm surges as we discuss in section 4.4, but there are other reasons for the small non-tidal residuals, apart from the resolution of GEBCO, including the resolution of the wind fields and the steep coastal topography, meaning there is only a limited near-shore shallow water zone. The resolution of the regional model is limited to ~12km (as stated in the manuscript), which means this nearshore zone is hardly resolved at the regional scale.*

2. In the same way, how can the authors be sure that the changes in the non-tidal residuals (section 4.4) during Hurricane Tomas and Ivan were mainly caused by the inverse barometer effect and not due to the model's low resolution (wind+bathymetry)?

Comparisons with measurements are too qualitative. The buoys used for validation seem to be very far from the cyclone tracks. Could the authors please comment on this?

*Reply*

*Much of the **modelled** non-tidal residuals appear to have been generated by the inverse barometer. Visually, this can seen by the shape of the non-tidal residual matching the low pressure area and changing very little with changes in ocean depth. However, some additional experiments were performed **(but not included in this study )** which omitted wind stress. In these experiments, there were some small changes to non-tidal residuals in grid cells very close to islands, however the order of magnitude of the surge remained the same and conclusions unchanged.*

*We have used the existing wave buoy network in the Caribbean, which shows good agreement at the regional scale. Tide gauge data is more problematic, because of the very local scale of the signal, as discussed in 4.4. Limited observations is another point raised in this manuscript (e.g., Line 44, 204, 244, 455, 460, 461). Our aim is to highlight the importance of observations in this publication while supporting local capability through the provision of an AWAC in SVG (see wolf et al., Deployment of an AWAC off the east coast of St Vincent, 2018-2019, http://nora.nerc.ac.uk/id/eprint/525933/)*

3. Line 175: the radius of maximum winds for Ivan could be estimated from the HURDAT Re-analysis dataset (https://www.aoml.noaa.gov/hrd/hurdat/Data_Storm.html)

*Reply*

*We have been using the Holland parametric model to build the wind and pressure field, which requires as input the along track observations of maximum wind, central pressure and radius of maximum wind. The latter was not available in the IBTRACS dataset for hurricane Ivan. The IBTRACS dataset is a merge of data from different data centers including HURDAT. We have further checked directly on the HURDAT2 website and the radius of maximum wind speed is not a variable provided. HURDAT2 provides instead the 34 kt wind radii maximum extent, 50 kt wind radii maximum extent and 64 kt wind radii maximum extent.*

4. How are the inputs from rivers and rainfall incorporated into the model grid? (boundary conditions?). What about other factors that contribute to total water levels such as vertical land movement (tectonic/seismic activity and anthropic) and wind waves (setup and runup) ? Could the authors provide more details/discussions on that? What are the consequences on their results of not considering these important processes? The literature review seems poor, I suggest to enlarge it looking at the recent advances in this region.

*Reply*

*Discussion is rewritten  (lines 486-510) to address these comments.*

*No rainfall or runoff data are included at this stage in our work. While these effects can be important for impacts in the coastal zone of SIDS e.g. causing combined flooding and landslides, these are not marine hazards, although they can combine with marine hazards to produce multi-hazard impacts of hurricanes, which are of course very important for coastal communities. No VLM data have been included. We intend to include calculations of wave runup, setup, erosion and geomorphology in a paper looking at the local scale for St Vincent, to be submitted shortly. We have included the suggested papers in the literature review, while clarifying the distinction between external drivers and local scale effects a bit more in section 5. We have noted the criticism of the literature review in general and have included several more papers, including those suggested by the reviewer and others.*

5. In addition, there is no scientific comparison/discussion in respect of recent storm surge research.
   Among others: Zahibo, N., Pelinovsky, E., Talipova, T., Rabinovich, A., Kurkin, A., & Nikolkina, I. (2007). Statistical analysis of cyclone hazard for Guadeloupe, Lesser Antilles. Atmospheric research, 84(1), 13-29.

   Krien, Y., Dudon, B., Roger, J., & Zahibo, N. (2015). Probabilistic hurricane-induced storm surge hazard assessment in Guadeloupe, Lesser Antilles. Natural Hazards & Earth System Sciences Discussions, 3(1).

   Krien, Y., Dudon, B., Roger, J., Arnaud, G., & Zahibo, N. (2017). Assessing storm surge hazard and impact of sea level rise in the Lesser Antilles case study of Martinique. Natural Hazards & Earth System Sciences, 17(9).

   Kennedy, A. B., Gravois, U., Zachry, B. C., Westerink, J. J., Hope, M. E., Dietrich, J. C., ... & Dean, R. G. (2011). Origin of the Hurricane Ike forerunner surge. Geophysical Research Letters, 38(8).

*Reply*

*We have added a few sentences in our text, e.g. lines 75-80, 334, 340, 465 and 487-526 to address these comments.*

*We thank the reviewer for pointing out these other studies and have included them in the literature review. Again, these are mainly local scale studies for the islands of Martinique and Guadeloupe and also for the Louisiana/Texas coast, which is a different situation, on the USA continental shelf. As previously stated, we have concentrated on assessing the external drivers at the regional scale, and focussing on SIDS, while recognising that the local response is important in terms of the impacts. Note that one of our points is the small size of surges in steep islands like Guadeloupe. Referring to the work of Zahibo et al. (2007), although observed surges are stated to be quite large, there is also a lot of variability and the modelled surge in tropical storm Lili (2002), although for a relatively low intensity event, was only about 10cm. We think some of the reported historical water levels may be due to waves and part is no doubt due to limited spatial resolution of the coastal bathymetry. Krien et al (2015; 2017) show 1 in 100y surges of up to 2m and 1 in 1000y surges up to 3m in the shallow water areas around Martinique and Guadeloupe, which are similar to our estimates, for an estimated severe event near St Vincent, allowing for our limited spatial resolution.*

6.  The vulnerability definition that is used by the authors remains vague. Please, elaborate more on the concepts of risk, exposure, vulnerability... In the last section, a Coastal Vulnerability Index (CVI), based on the methodology used by Thieler and HammarKlose (1999) is presented, gathering six variables : geomorphology; coastal slope; shorline erosion/accretion/ relative slr; mean tide and mean wave.
    However, geomorphology is not considered in the study, slr is constant for all the study region (1.8mm/yr), no justification/references is given for the choice of the shoreline changes (between - 1.0 and +1.0m yr-1 ), so only mean weight height and tidal range provide relevant information to compute the CVI. The same observation is made concerning the choice of the CVIPP from OE210, where the human interventions at the coast and the coastal geomorphological factors are ignored. At the end, what is the meaning of this vulnerability index (figure 11)?

*Reply*

*Session 5 has been improved following reviewers' comments*

*In section 5 we initially stated that we have adapted the CVI method of Thieler and Hammar-Klose (1999), in other words we have used the concept but not exactly followed the methodology. We have added some clarification to the section on Coastal Vulnerability Index, indicating the difference between external physical drivers (addressed here) and local coastal variables including geomorphology, beach slope and rate of retreat, for which we do not have data on the whole Caribbean scale. Some attempts could be used to get these data from a global model like DIVA (Dynamic Interactive Vulnerability Analysis, although the data would need to be check to ensure the islands are resolved), but this was not the purpose of the present paper, which was to quantify the exposure to marine hazards, rather than the local response. We will quantify these local variables in a second paper for St Vincent.*

*We have modified the text at line 400 to clarify that sea level rise and erosion rates applied are defined by the approach of THK. We use these as thresholds to rank the variable rates of sea level and shoreline change in our study (Table 6).*

Minor comments:

a)  The figures 1 and 11 are not well constructed neither clear for understanding. For example, in Figure 1 : the names of countries and the names tide gauge stations are missing...

*Reply: Done*

b)  L217 : please correct : Is the comparison made with Pointe-à-Pitre tide gauge located in Guadeloupe or with the Port au Prince tide gauge located in Haiti ?

*Reply:*

*Pointe-a-Pitre in Guadeloupe*

Quantifying processes contributing to coastal hazards to inform marine climate resilience assessments, demonstrated for the Caribbean Sea

MS No.: nhess-2020-46

Reply to Reviewer 2 comments

The manuscript details a methodology for regional assessment of coastal impacts due to waves, storm surges and sea level rise. The study focuses on Caribbean SIDS (Small Island Developing States), which are particularly vulnerable to coastal climate change, since they strongly rely on the preservation of the coastal zone and they are also prone to natural disasters. The adopted approach can provide information for the design and implementation of the requisite coastal adaptation strategies. I recommend the publication of the manuscript following some clarifications and minor corrections.

1. The terms risk, exposure and vulnerability are used in a confusing way (e.g. Pg. 1 lines 18-19: "We introduce a Combined Vulnerability Index, which allows a quantitative assessment of relative risk across the region, showing that sea level rise is the most important risk factor everywhere" and Pg. 13 lines 404-405: "we can calculate an external physical exposure factor, including the rate of sea level rise, the wave climate and tidal range, which we here refer to as a Combined Vulnerability Index (CBVI)"). Please clarify better these concepts.

*Reply: Section 5 has been modified to clarify the terms and concepts.*

2. An analysis was made regarding hurricane induced storm surges and a Combined Vulnerability Index (CBVI) is proposed for marine hazards; however storm surges are not included on the proposed CBVI. Could the authors comment on that?

*Reply: Text in chapter 5 and Discussion has been modified. Lines ****

*The vulnerability index uses tidal range as the variable to describe the vulnerability of the coastline to time-varying water level. In a microtidal area like in most of the Caribbean this makes most of the coastlines vulnerable to surges. We added a wind-speed variable into the CVI in order to allow for the occurrence of hurricanes, which will have direct and indirect (surge) impacts.*

3. The names of the countries and the locations mentioned in section 2 should be added in Figure 1.

*Reply: As suggested Figure 1 has been updated to include the country names.*

4. Figure 5 does not include very important information for the manuscript; it could be transferred in the supplementary material or it could be combined with figure 6.

*Reply: Following the suggestion from Reviewer 1 Figure 5 has been moved into SI.*

5. The caption in Figure 7 (Pg.27, line 686) mentions "Locations of wave buoys used for model validation are marked as filled red circles" but the locations are missing from the

figure. The map in Figure 7 looks deformed, the scale of the figure should be corrected.

*Reply: Figure caption has been corrected.*

6. Caption of figure 8 (Pg. 28, lines 696-697): It is not clear that Fig. 8c and 8d concern Hurricane Tomas.

*Reply: As suggested the caption to Figure 8 has been updated to improve clarity.*

7. Figure 11 is not very clear, needs to be improved.

*Reply: This figure 11 (figure 10 in our updated manuscript) has been modified to improve clarity as requested.*

8. Figure SI1 and Table SI1 of the supplementary material is not mentioned in the manuscript.

*Reply: Table SI1 is now mentioned in Line 107*

9. Pg. 9, lines 250-251: "Future sea level projections for RCP8.5, including the low probability/ high impact scenario (the 95th percentile) are shown on Figure SI4". It's Figure SI5 not Figure SI4.

*Reply: Done*

10. Pg. 10, line 307: "Fig. 11 show maximum non-tidal residual envelopes for the case study events discussed in Section 3.3.1". It's Fig. 10 that depicts the non-tidal residual envelopes, not Fig. 11.

*Reply: Done*

11. Caption of figure 6 (Pg. 26, line 680): Please correct the typing error "(whereas the global mean trend is 3.00+/-0.4mm yr-1 is removed)"

*Reply: Done*

**Quantifying processes contributing to  marine hazards to inform coastal climate resilience assessments, demonstrated for the Caribbean Sea**

5    S. Jevrejeva [1,2], L. Bricheno[1], J. Brown[1], D. Byrne[1], M. De Dominicis[1], A. Matthews[1], S. Rynders[1], H. Palanisamy[2] and J. 
[revised manuscript text omitted]

75        Various previous studies have been carried out for islands in the Lesser Antilles, especially those belonging to the wealthier nations of Netherlands and France, such as Martinique, Guadeloupe and Saint-Martin (Zahibo et al., 2007; Krien et al., 2015; 2017). These are not strictly SIDS, because they have the resources of developed nations to support them. Another study, by Kennedy et al. (2011), explores the mechanism of the storm surge on the Louisiana/Texas continental shelf around the USA, whereas we are mostly looking at steep volcanic islands, with a very narrow continental shelf.

[revised manuscript text omitted]

**5 Combined  Vulnerability Index**

350 Coastal threats from the sea can include flooding and coastal erosion caused by steadily rising sea levels as well as changes on other time-scales e.g. changing tides, storm surges and waves. Here we address the challenge of combining these different processes into a single vulnerability index, which can allow us to compare different types of coastline on a national, regional and global scale. In Sutherland and Wolf (2002) qualitative and quantitative differences in future changes in coastal vulnerability were found between five sites selected around the coastline of England and Wales, for the present-day

355 and an estimate of the future climate in 2075. The differences arose because the sites have different tidal ranges, wave climates and surge levels. Moreover, the parameters have different joint probabilities at different sites. Thus results from one site cannot be transferred directly to other sites and individual assessments must be made for specific sites.

Coastal vulnerability may be defined as the degree to which a system is susceptible to, or unable to cope with, adverse effects of climate change, including climate variability and extremes. In IPCC (2012), vulnerability is defined as the propensity

360 or predisposition to be adversely affected. This may result from a combination of external hazards and local conditions.

An accurate and quantitative approach to predicting coastal change is difficult to establish. Even the kinds of data necessary to predict shoreline response are the subject of scientific debate. A number of predictive approaches have been proposed (National Research Council, 1990 and 1995), including:

- extrapolation of historical data (e.g., coastal erosion rates),
365 - static inundation modelling,
- application of a simple geometric model (e.g., the Bruun Rule),
- application of a sediment dynamics/budget model, or
- Monte Carlo (probabilistic) simulation based on parameterized physical forcing variables.

However, each of these approaches has inadequacies or can be invalid for certain applications (National Research Council,

370 1990). Additionally, shoreline response to sea-level change is further complicated by human modification of the natural coast such as beach nourishment projects, and engineered structures such as seawalls, revetments, groins, and jetties. Understanding how a natural or modified coast will respond to sea-level change is essential to preserving vulnerable coastal resources.

Here we adapt the Coastal Vulnerability Index (CVI), derived by Thieler and Hammar-Klose (1999, hereafter THK), by dividing it into the  local physical variables (geomorphology, shoreline erosion/accretion rate and coastal slope)

375 and the external physical variables (relative SLR rate, mean significant wave height and mean tidal range). The local variables require specific knowledge of the coastal typology (which can be calculated for detailed coastal studies on a local scale), whereas the  external variables can be seen rather as external drivers and can be quantified from the regional models described above.

THK define the *CVI* as a combination of  the local and external physical variables as follows (with a slight re-definition of parameters:

$$CVI = \sqrt{\frac{(a * b * c) * (d * e * f)}{6}}$$

where,

  *a* = geomorphology factor

  *b* = coastal slope factor

  *c* = shoreline erosion/accretion rate

  *d* = relative sea-level rise rate factor

  *e* = mean tide range factor

  *f* = mean wave height factor.

Following  THK we may consider *a,b,c*  to be local/geomorphological parameters: and *d,e,f* are external physical parameters.

Table 1 shows the six variables described by THK, which include both quantitative and qualitative information. The five quantitative variables are assigned a vulnerability ranking based on their actual values, whereas the non-numerical geomorphology variable is ranked qualitatively according to the relative resistance of a given landform to erosion. Shorelines with erosion/accretion rates between -1.0 and +1.0m yr$^{-1}$ are ranked as being of moderate vulnerability in terms of that particular variable. Increasingly higher erosion or accretion rates are ranked as correspondingly higher or lower vulnerability. Regional coastal slopes range from very high vulnerability, <4.59 percent, to very low vulnerability at values >14.7 percent. The rate of relative sea-level change is ranked using the rate of global sea level over the 20$^{th}$ century (1.8mm yr$^{-1}$) as very low vulnerability. Since this is a global or "background" rate common to all shorelines, the sea-level rise ranking reflects primarily local to regional isostatic or tectonic adjustment. Mean wave height contributions to vulnerability range from very low (<1.1 m) to very high (>2.6 m). Tidal range is ranked such that micro-tidal (<1m tidal range) coasts are very high vulnerability and macro-tidal (>6m tidal range) coasts are very low vulnerability.

There have been various developments of the indicator-based approach e.g. Ramieri et al (2011). New models have been developed for the assessment of coastal vulnerability at the global to national level e.g. the DIVA model (Hinkel and Klein, 2009).

Following Özyurt and Ergin (2010, hereafter OE2010) an additive model has been used to combine parameters, with normalisation, which allows more quantitative inter-comparison between different coastline, with respect to exposure and resilience, so we have used this rather than the multiplicative combination in THK. OE2010 added further physical parameters,

representing hydrological parameters, such as groundwater and river discharge, and also add a set of human parameters, representing anthropogenic changes, such as hard defences, reduction in sediment supply and land use. This divides the model into 2 subsets:

415    $CVI = (\frac{1}{2}\,CVIPP + \frac{1}{2}CVIHP)/CVILV$, where $CVILV$ is the least vulnerable location, used as a normalisation factor.

Here $CVIPP$ is derived from the physical parameters and $CVIHP$ from the human parameters, such that
$CVIPP = \sum PP_n.R_n$ (summed over all $n$ physical factors considered, $PP_n$, with weighting factors $R_n$) and $CVIHP = \sum HP_m.R_m$ (summed over all $m$ human factors considered, $HP_m$, with weighting factors $R_m$).

420    If we ignore the human interventions at the coast and the coastal geomorphological factors (which we have not derived at the regional scale), we can calculate an external physical exposure factor, including the rate of sea level rise, the wave climate and tidal range, which we here refer to as a Combined Vulnerability Index (*CBVI*), which refers to the exposure of the coast to these external factors. We have calculated the vulnerability indices, for each factor, for all coastal points around the Caribbean, using the data derived in sections 3 and 4 and the parameter ranges in Table 1. As the tide is microtidal, the tidal

425    vulnerability index is a maximum nearly everywhere. Likewise, there is a saturation of the index for the rate of sea level rise, because the present-day rate of 2.8 mm yr$^{-1}$ exceeds the maximum in the original definition everywhere and future projections lead to much higher rates. There is some variation of the wave index, showing larger vulnerability at more exposed coasts. We have also chosen to include an additional storminess index as most of the Caribbean is microtidal, which makes the coastline vulnerable to surges. A wind-speed variable is used in order to allow for the occurrence of hurricanes, which will have direct

430    and indirect (surge and wave) coastal impacts., tTo represent the amount of exposure to severe storms: the data are used for this isfrom a 30- year climatology, taken from the ERA5 reanalysis dataset. From this, we have chosen the mean annual maximum wind-speed (*WSMAM*) as the parameter which best illustrates the variation in storminess across the region. The range (1-5) has been set as follows: *WSMAM*) <10m s$^{-1}$, index = 1; 10< *WSMAM* <15, index = 2; 15< *WSMAM* <20, index = 3; 20< *WSMAM* <25, index = 4; -*WSMAM* >25m s$^{-1}$, index = 5. The more southerly coastlines, close to South America, have

435    the lowest vulnerability to storm winds, then the vulnerability increases northward along the Windward and Leeward Islands, reaching a maximum around Anguilla. The Gulf of Mexico shows generally higher vulnerability and the Turks and Caicos Islands (outside the Caribbean Island arc), are particularly susceptible to strong winds, being along the main tropical storm track. See Fig. 11a 10a for the variation of the *CBVI* without winds, over the Caribbean, while Fig. 11b 10b shows the *CBVI* including winds. Note that, for the present, a uniform weighting factor (i.e. 1) is has been applied to all the 4 indices, in the

440    absence of a better way of assessing what this should be. When wind is included in the *CBVI*, this variable has a large dynamic range, with a few hotspots where the value reaches 5, but has the effect of reducing the *CBVI* over less exposed coastlines.

**6    Discussion**

In this study we have quantified individual processes contributing to coastal hazards in the Caribbean, suggesting that in the present-day climate the greatest hazards at the coast are is due to waves, with observed and modelled height for extreme events up to 5 m near the coast and maximum of 12 m offshore. Note that the modelled waves are expected to be underestimated during the most extreme events, since winds are under-estimated in the atmospheric model, due to limited spatial resolution, particularly for the most extreme events. For the synoptic view, a reanalysis data set, ERA5, covers the full range of conditions across the Caribbean Sea. Due to the coarseness of the model grid, some small islands are unresolved, however the missing land is represented in the model by a partial obstruction (Tolman, 2009). A sheltering effect can be seen around the Windward Islands (62° W, 10 - 16° N), even though the archipelago is not explicitly present as 'land points'.

We have simulated storm surges for the past extreme events associated with Hurricane Ivan (2004) and Hurricane Tomas (2010), providing estimates of storm surge up to 0.16m in the coastal areas of SVG. In aAn additional scenario, using a hurricane that represents a category 4 event on a direct trajectory for SVG, we simulated storm surge which might reach up to 1.5m for a direct hurricane impact, depending on the resolution of the underwater topography. Tidal range for most of locations is less than 0.5m, due to deep water close to shore. The tide can be amplified in over the shallower continental shelves e.g. east of Nicaragua and around Trinidad.

The regional sea level trend, calculated from satellite altimetry data for the period 1993-2018, is $2.8\pm 0.4$mm yr$^{-1}$ compared to the global mean trend of $3.0\pm 0.4$mm yr$^{-1}$, which is in a good agreement from previous studies with shorter time period (Torres and Tsimplis, 2013; Palanisamy et al., 2012). Several estimates from tide gauge records provide a wide range (0.3-12mm yr$^{-1}$) of past sea level rise rates for individual locations with different observational time period (Torres and Tsimplis, 2013; Palanisamy et al., 2012). The large difference in rate for individual locations is explained by the lack of consistent observations, gaps in time series, and lack of information about the vertical land movement in location of tide gauge (Holgate et al., 2003; Palanisamy et al., 2012; Torres et al., 2013).

The main limitations to estimating present day processes contributing to coastal hazards are:

- Lack of the observational data for assessment of current changes in sea level, storm surges and waves. In addition, this lack of observations is limitings the model calibration. It seems there is a relation between model performance and the number of gaps in the observations. The best fits are for continuous records, the ones with big gaps do not do so well.

-

- Lack of sufficient forcing (e.g. local wind data) to simulate extreme events at the coast is a challenge to reproduce past events. Long reanalyses do allow us to examine extreme events over the last nearly 200 years, e.g. NOAA-CIRES-DOE Twentieth Century Reanalysis (V3), but there is still an issue with insufficient spatial resolution (typically 1° latitude/longitude) to capture the true intensity of hurricane winds.

- We present a vulnerability metric for the region, allowing intercomparison of the risks from marine hazards in different locations. However, we notice that this approach needs to be  refined, taking into account the larger changes in physical variables (e.specially relative SLR rate) for different future climate scenarios.

- In this study we have quantified individual processes contributing to coastal hazards on the regional scale, for the Caribbean. We have not investigated the interaction between these processes, e.g. between water level, waves and storm surges. In general these interaction effects are secondary, at a regional scale, but may be more significant at a local scale. For steep volcanic islands, large waves are often the dominant contributor to increased water levels during storms, as deep water close to shore permits large waves to reach the coast while limiting the generation of storm surges.

- Some issues are not addressed here, such as coastal inundation, including wave runup, which requires detailed knowledge of the local coastal topography. Some preliminary work on this has been carried out for St Vincent, using reconstructed beach profiles (Prime et al., 2019) and this will be addressed in future work.

The Combined Vulnerability Index calculated on the base of current climate conditions demonstrates that islands on the eastern  side of the basin are more vulnerable, due to impact from wind and waves. However, with an increasing rate of sea level rise by 2100 for all emission scenarios, sea level rise is imposing a large hazard in the region. Extreme sea levels associated with hurricanes, tropical cyclones and waves are projected to change in a warming climate. Nevertheless, recent results from Morim et al. (2019) for wave projections under future climate scenarios for the eastern Caribbean Sea (for the more extreme 'business-as-usual' RCP8.5 emissions scenario) shows no change or a slight reduction (<5%) in mean and extreme wave height. When divided into seasons there can be seen a small increase (<5%) in mean wave height during the height of the summer (JJA), which also coincides with the hurricane season.

Coastal impacts on each island are governed by their coastal morphology, which is not analysed in the present paper: here we are focussing on the exposure of the whole Caribbean Sea coastline to marine hazards rather than  examining the detailed impacts for individual islands (although we do refer to the case study for SVG to illustrate local impacts as a consequence of regional processes). This  present s a challenge, particularly for estimating coastal storm surges, as we discuss in section 4.4, but there are other reasons for the small non-tidal residuals, apart from the resolution of GEBCO, which include the resolution of the wind fields and the steep coastal topography, meaning there is only a limited near-shore shallow water zone. The resolution of the regional model is limited to ~12km, which means this nearshore zone is hardly resolved at the regional scale. In addition, no rainfall or runoff data are included at this stage in our work. While these effects can be important for impacts in the coastal zone of SIDS, e.g. causing combined flooding and landslides, these are not marine hazards, although they can combine with marine hazards to produce multi-hazard impacts of hurricanes, which are of course very important for coastal communities. Vertical land movement, seismic and anthropogenic activities have not been considered in this study due to lack of observational data in the islands; however, as previous studies suggest (e.g. Dasgupta et al, 2009; CARIBSAVE, 2012) an impact of climate change on the coast is expected to be altered if non-climate conditions are included.

**Commented [JS1]:** Maybe we could add a sentence regarding some further development of the CVI here or at the end of previous chapter (CVI chapter).

In our study we have demonstrated that a storm surge might reach 1.5m for a direct hurricane impact depending on the underwater topography under an additional scenario hurricane that represents a category 4 event. We have demonstrated that a storm surge might reach 1.5m for a direct hurricane impact depending on the underwater topography under an additional scenario hurricane that represents a category 4 event. This is in a good agreement with previous modelled surges up to 1m (return period 100 years) and up to 3m (return period 1000 years) in the shallow water areas around Martinique and Guadeloupe during the hurricane activitys (Krien et al., 2015; 2017). However, the small size of storm surges in the simulations in our study for the Caribbean region areis supported by the work by Zahibo et al. (2007); although observed surges are stated to be quite large, there is also a lot of variability and the modelled surge in tropical storm Lili (2002), althoughbeit for a relatively low intensity event, was only about 10cm. Projections for the 21st century indicate that it is likely that the global frequency of tropical cyclones will either decrease or remain essentially unchanged, along with a likely increase in both global mean tropical cyclone maximum wind speed and rain rates (IPCC, 2013). The frequency of the most intense storms will more likely than not increase in some basins. More extreme precipitation is projected near the centres of tropical cyclones making landfall in North and Central America (IPCC, 2013).

A comparative study of the impact of sea level rise on coastal inundation across 84 developing countries showed that the greatest vulnerability to a 1m sea level rise in terms of inundation of land area was located in East Asia and the Pacific, followed by South Asia, Latin America, and the Caribbean, the Middle East and North Africa, and finally sub-Saharan Africa (Dasgupta et al., 2009). Future sea level rise will pose a significant threat to the coastal infrastructure, settlements, beaches, ecosystems and economic activity in coastal areas of SIDs. In St Vincent, sea level rise of 1m will place 10% of the main tourist infrastructure at risk, along with 67% of sea port infrastructure (CARIBSAVE, 2012). In addition, sea level rise will alter affect the erosion of the coastline, with approximately 100 m of erosion resulted from 1 m of sea level rise (CARIBSAVE, 2012). Up to 70% of the major tourist resorts will be impacted by the disappearance of the beaches due to sea level rise and erosion (CARIBSAVE, 2012).

Here, we focus on assessing the external drivers, of sea level rise, tides, storm surges and waves as well as winds, showing the relative exposure of SVG to these factors, compared to other parts of the Caribbean Sea. A following paper will assess the local impacts on the island of St Vincent (Wolf et al., in preparation). Further work is needed to calculate coastal erosion, flooding and economic impacts in SIDS e.g. Villaroel-Lamb (2020) has shown how a coastal process model, driven by offshore wave and surge conditions, based on the Saffir-Simpson scale (method prior to 2010: Walker et al., 2018 discuss the limitations of this method) can be used to estimate rates of coastal erosion and economic loss, using generic coastal structures, for Caribbean beaches.

**7      Conclusion**

For Caribbean island nations it is crucial that the spatial variability in hazardous wave and water level conditions and the contributions of different processes in causing hazardous conditions are understood, in order to prepare and resource island-

540 wide coastal climate resilience plans. In our study, we have explored the role of waves, storm surges and sea level rise for the Caribbean region with a focus on coastal impacts in the eastern Caribbean, using St Vincent and the Grenadines (SVG) as an example. We identified that currently the greatest hazard is waves, with up to 12 m wave height offshore and 5 m near the coast for extreme events (e.g. Hurricane Ivan/Tomas), which can lead to overtopping, runup, wave setup and coastal erosion. For the same extreme events the storm surge height may be up toreach 1.5m. Sea level rise since 1993 is about 0.1m, which is
545 close to the global sea level rise estimate for the same time period. However, sea level rise is accelerating and by 2100 projected sea level rise could exceed 2m by 2100 with RCP8.5 scenario.

[revised manuscript text omitted]

Melet., A., Almar, R. and Meyssignac, B.: What dominates sea level at the coast: a case study for the Gulf of Guinea, Ocean
660 Dynamics, 66:623–636, 2016.

Melet., A., Meyssignac, B., Almar, R. and Le Cozannet, G.: Under-estimated wave contribution to coastal sea-level rise, Nature Climate Change, 8, 234- 239, 2018

Merrifield, M. A., Becker, J.M., Ford, M. and Yao, Y.: Observations and estimates of wave-driven water level extremes at the Marshall Islands, Geophysical Research Letters, 41, 7245–7253, 2014.

665 Monioudi et al.: Climate change impacts on critical international transportation assets of Caribbean Small Island Developing States (SIDS): the case of Jamaica and Saint Lucia. Reg Environ Change 18, 2211–2225, 2018.

Morim, J., Hemer, M., Wang, X.L., Cartwright, N., Trenham, C., Semedo, A., Young, I., Bricheno, L., Camus, P., Casas-Prat, M., Erikson, L., Mentaschi, L., Mori, N., Shimura, T., Timmermans, B., Aarnes, O., Breivik, Ø., Behrens, A., Dobrynin, M., Menendez, M., Staneva, J., Wehner, M., Wolf, J., Kamranzad, B., Webb, A., Stopa, J. and Andutta, F.:
670 Robustness and uncertainties in global multivariate wind-wave climate projections, Nature Climate Change, 9, 711–718, 2019.

Özyurt, G. and Ergin, A.: Improving coastal vulnerability assessments to sea-level rise: a new indicator based methodology for decision makers, Journal of Coastal Research, 26(2), 265–273, 2010.

675    Palanisamy, H., M. Becker, B. Meyssignac, O. Henry:  Regional sea level change and variability in the Caribbean Sea since 1950. Journal of Geodetic Science, 2, 2125–2133, 2012.

Pielke Jr., R.A., Rubiera, J., Landsea, C., Fernandez, C.M.L. and Klein, R.: Hurricane vulnerability in Latin America and the Caribbean: normalized damage and loss potentials. Natural Hazards Review, 4 (3), 101–113, 2003.

680    Peltier, W R, Argus, D. F. and Drummond, R.: Space geodesy constrains ice age terminal deglaciation: the global ICE-6 G_C (VM5a) model, J. Geophys. Res. Solid Earth, 450–87, 2015.

Pendleton, E.A., Thieler, E.R. and Williams, S.J.: Coastal Vulnerability Assessment of Virgin Islands National Park (VIIS) to Sea-Level Rise. U.S. Geological Survey Open-File Report 2004-1398, https://pubs.usgs.gov/of/2004/1398/, 2005.

Prime, T., Brown, J. and Wolf, J.: St Vincent – Black Point Beach Modelling. National Oceanography Centre Research and
685    Consultancy Report, no. 70. National Oceanography Centre, UK, 2019.

Rhiney,  K.  Geographies of Caribbean vulnerability in a changing climate: issues and trends. Geography Compass 9(3):97–114. https://doi.org/10.1111/gec3.12199, 2015.

Sutherland and Wolf.: Coastal Defence Vulnerability 2075. Hydraulics Research Scientific Report SR590, 220pp, 2002.

690    Splinter, K.D., Carley, J., Golshani, A. and Tomlinson, R.B.: A relationship to describe the cumulative impact of storm clusters on beach erosion. Coastal Engineering, 83, 49-55, 2014.

Splinter, K., Davidson, M., Golshani, A. and Tomlinson, R.B.: Climate controls on longshore sediment transport, Continental Shelf Research, 48, 14-156, 2012.

Splinter, K.D., Palmsten, M. L, Holman, R. A. and Tomlinson, R. B.: Comparison of measured and modeled run-up and
695    resulting dune erosion during a lab experiment, Proceedings of Coastal Sediments, Miami, FL, USA: World Scientific, 782-795, 2011.

Splinter, K., Strauss, D. and Tomlinson, R.: Can we reliably estimate dune erosion without knowing pre-storm bathymetry? P. C. Pattiaratchi (Ed.), Proceedings of Coasts and Ports, 2011, Perth, Australia.

Stockdon, H. F., Holman, R. A., Howd, P. A. and Sallenger, A. H.: Empirical parameterization of setup, swash, and runup,
700    Coastal Engineering, 53, 573–588, 2006.

Stockdon, H.F., Sallenger Jr., A. H., Holman, R. A. and Howd, P. A.: A simple model for the spatially variable coastal response to hurricanes. Marine Geology, 238, 1-20, 2007.

Thieler, E.R., and Hammar-Klose, E.S.: National assessment of coastal vulnerability to sea-level rise, U.S. Atlantic Coast: U.S. Geological Survey Open-File Report 99-593, 1999. https://pubs.usgs.gov/of/1999/of99-593/

705    Tolman, H.: User manual and system documentation of WAVEWATCH III[tm]-version 3.14 (Tech. rep.): NOAA / NWS / NCEP / MMAB Technical Note-276, 220, 2009.

Torres, R.R, and Tsimplis, M.N.: Sea-level trends and interannual variability in the Caribbean Sea. J Geophys Res 118:2934–2947. https://doi.org/10.1002/jgrc.20229, 2013.

United Nations, 2015. SMALL ISLAND DEVELOPING STATES IN NUMBERS CLIMATE CHANGE EDITION 2015, available from http://unohrlls.org/sids-in-numbers-climate-change-edition-2015/.

van de Wal RSW et al. : Uncertainties in long-term twenty-first century process-based coastal sea-level projections. *Surv Geophys*. https://doi.org/10.1007/s10712-019-09575-3, 2019.

Villarroel-Lamb, D. : Quantitative Risk Assessment of Coastal Erosion in the Caribbean Region. Natural Hazards Review, 31, 3, doi: 10.1061/(ASCE)NH.1527-6996.0000388, 2020.

Wada Y, et al.,: Past and future contribution of global groundwater depletion to sea-level rise, Geophys Res Lett., **39**(9), 2012.

Williams S.J. et al.: Physical Climate Forces. In: Burkett V., Davidson M. (eds) Coastal Impacts, Adaptation, and Vulnerabilities. NCA Regional Input Reports. Island Press, Washington, DC, 2012.

Wolf, J.: Practical aspects of physical oceanography for small island states, pp. 120-131 in 'Small Islands: marine science and sustainable development' ed. George Maul, AGU Coastal and Estuarine Studies Series, 1996

Wolf, J. and Woolf, D. K.: Waves and climate change in the north-east Atlantic, Geophysical Research Letters, 33, L06604, doi:10.1029/2005GL025113, 2006.

Zahibo, N., Pelinovsky, E., Talipova, T., Rabinovich, A., Kurkin, A. and Nikolkina, I. :  Statistical analysis of cyclone hazard for Guadeloupe, Lesser Antilles. Atmospheric Research, 84(1), 13-29, 2007.

Table 1: Ranges for Vulnerability Ranking of Variables on the Atlantic/Caribbean Coast (from THK)

| Variables | Very Low 1 | Low 2 | Moderate 3 | High 4 | Very High 5 |
|---|---|---|---|---|---|
| Geomorphology | Rocky cliffed coasts, Fjords | Medium cliffs, Indented coasts | Low cliffs, Glacial drift, Alluvial plains | Cobble Beaches, Estuary, Lagoon | Barrier beaches, Sand beaches, Salt marsh, Mud flats, Deltas, Mangrove, Coral reefs |
| Shoreline erosion/accretion (m yr$^{-1}$) | > 2.0 | 1.0 - 2.0 | -1.0 - 1.0 | -2.0 - -1.0 | < -2.0 |
| Coastal slope (%) | > 14.70 | 10.90 - 14.69 | 7.75 - 10.89 | 4.60 - 7.74 | < 4.59 |
| Rate of sea level changes (mm yr$^{-1}$) | < 1.8 | 1.8 - 2.5 | 2.5 - 3.0 | 3.0 - 3.4 | > 3.4 |
| Mean wave height (m) | < 1.1 | 1.1 - 2.0 | 2.0 - 2.25 | 2.25 - 2.60 | > 2.6 |
| Mean tide range (m) | > 6.0 | 4.0 - 6.0 | 2.0 - 4.0 | 1.0 - 2.0 | < 1.0 |

730

[Figure]

[Figure]

735 **Figure 1: Map of the Caribbean Sea, showing bathymetry and locations of 12 tide gauges and 7 wave buoys used for models validation. Tables SI2 and SI3 give the locations of the tide gauges and wave buoys respectively.**

[Figure]

**Figure 2: Time series showing a zoom to the dates of Hurricane Tomas (Oct 2010) from the global model and**
740 **observations at the buoy 42060 (see Fig. 1 for location).**

[Figure]

**Figure 3: Modelled and observed non-tidal residuals (m) during the passing of Hurricane Maria (2017) over the Port-au-Prince tide gauge (Port-au-Prince, Guadeloupe).**

745

[Figure]

**Figure 4: Comparison between the significant wave heights observed by 5 buoys from the NOOA National Data Buoy Center and modelled by the WW3 Caribbean model forced by ERA5 reanalysis winds. Validation is for October 2010 and November 2010, covering the passage of Hurricane Tomas (29 October 2010 – 14 November 2010).**

750

[Figure]

**Figure 5: Sea level trends in the Caribbean basin** (whereas the global mean trend $3.00 \pm 0.4$ mm yr$^{-1}$ is removed), calculated using satellite altimetry data 1993-2018.

[Figure]

755

**Figure 6: Mean significant wave height (m, colour shading) and mean direction (vectors) from the global 37-year historic climatology wave model run.**

760

765

[Figure]

**Figure 7: Maximum significant wave height simulated by the WW3 Caribbean model during a) Hurricane Tomas (2010) and b) Hurricane Ivan (2004) forced by ERA5 and c) Hurricane Tomas (2010) forced by the Holland model parametric wind field and d) Hurricane Tomas (2010) forced by the enhanced wind field.**

[Figure]

775

**Figure 8: Significant wave height at Argyle airport during Hurricane Tomas: red line – wave model run forced with the ERA5 reanalysis; blue line – wave model forced with the Holland's model parametric winds; pink line – wave model forced with "worst case scenario" enhanced winds.**

780

785

[Figure]

**Figure 9: Maximum non-tidal residual envelopes for four model runs: a) ERA5-forced Hurricane Tomas, b) ERA5-forced Hurricane Ivan, c) Holland-forced Hurricane Tomas and d) 'Enhanced Tomas' case study.**

790

[Figure]

[Figure]

b)

**Figure 10: Coastal vulnerability metrics; a) combined based on mean annual rate of sea level rise, mean wave height and Great Diurnal Tidal Range and b) as a) plus maximum windspeed**

795